# Genome-wide mutational biases fuel transcriptional diversity in the *Mycobacterium tuberculosis* complex

Álvaro Chiner-Oms [1,2,12], Michael Berney [3,12], Christine Boinett[4,5], Fernando González-Candelas [1,6], Douglas B. Young[7], Sebastien Gagneux[8,9], William R. Jacobs Jr[3], Julian Parkhill [10], Teresa Cortes [11] & Iñaki Comas [2,6]

The *Mycobacterium tuberculosis* complex (MTBC) members display different host-specificities and virulence phenotypes. Here, we have performed a comprehensive RNAseq and methylome analysis of the main clades of the MTBC and discovered unique transcriptional profiles. The majority of genes differentially expressed between the clades encode proteins involved in host interaction and metabolic functions. A significant fraction of changes in gene expression can be explained by positive selection on single mutations that either create or disrupt transcriptional start sites (TSS). Furthermore, we show that clinical strains have different methyltransferases inactivated and thus different methylation patterns. Under the tested conditions, differential methylation has a minor direct role on transcriptomic differences between strains. However, disruption of a methyltransferase in one clinical strain revealed important expression differences suggesting indirect mechanisms of expression regulation. Our study demonstrates that variation in transcriptional profiles are mainly due to TSS mutations and have likely evolved due to differences in host characteristics.

[1] Unidad Mixta "Infección y Salud Pública" FISABIO-CSISP/Universidad de Valencia, Instituto de Biología Integrativa de Sistemas-I2SysBio, Valencia, Spain. [2] Instituto de Biomedicina de Valencia, IBV-CSIC, Valencia, Spain. [3] Department of Microbiology and Immunology and Department of Molecular Genetics, Albert Einstein College of Medicine, New York, USA. [4] Sanger Institute, Wellcome Genome Campus, Hinxton, Cambridge, UK. [5] Hospital for Tropical Diseases, Wellcome Trust Major Overseas Programme, Oxford University Clinical Research Unit, Ho Chi Minh City, Vietnam. [6] CIBER en Epidemiología y Salud Pública, Valencia, Spain. [7] The Francis Crick Institute, London, UK. [8] Swiss Tropical and Public Health Institute, Basel, Switzerland. [9] University of Basel, Basel, Switzerland. [10] Department of Veterinary Medicine, University of Cambridge, Mandingley Road, Cambiddge CB3 OES, UK. [11] Department of Infection Biology, Faculty of Infectious and Tropical Diseases, London School of Hygiene and Tropical Medicine, London, UK. [12] These authors contributed equally: Álvaro Chiner-Oms, Michael Berney. Correspondence and requests for materials should be addressed to T.C. (email: Teresa.Cortes@lshtm.ac.uk) or to I.C. (email: icomas@ibv.csic.es)

In non-recombining bacteria, where mutation supplies most of the genetic variation, selective and non-selective processes can have a large impact on functional diversification[1]. Some of these functional differences can translate to phenotypic characteristics. This is the case for the *Mycobacterium tuberculosis* complex (MTBC), which despite its extremely low diversity, displays important biological differences between strains and phylogenetic lineages. For example, there are multiple examples of the association of MTBC lineages with specific populations[2,3] and in some settings this association could be linked to differential transmission efficacy depending on the host population[4,5]. Apart from transmission, the progression from latent infection to active disease differs among the different MTBC members[6]. Strains can also differ in other characteristics as for example growth rates in different in-vitro and in-vivo conditions, elicitation of immune responses or pathology in infection models[7]. Some of these phenotypic characteristics seem to depend on transcriptional differences[8,9], as illustrated by the gene expression differences reported in MTBC strains grown in-vitro and in-cellula[10,11].

Studies on a limited set of reference strains show that transcriptional differences in the MTBC can be mediated by differential action of transcriptional factors[12], methylation patterns[13] or expression of non-coding RNAs[14–16]. For some cases, the genetic bases of the expression differences are known. We have previously shown that MTBC regulatory networks vary across strains and lineages, with several transcription factors carrying mutations that potentially impair regulatory function[17]. In addition, major expression changes can be linked to sequence variants that affect coding regions of a signalling cascade[18] or create new transcriptional start sites (TSS)[19], especially if they affect regulatory hubs. Some of these new TSS have been previously reported to be favoured by a genome-wide mutational bias in the MTBC towards AT genetic changes[20,21]. However, the link between individual variants, underlying population processes and phylogeny-wide transcriptional diversity is still missing.

To date, gene expression studies (mainly based on microarray technology and some on RNA-seq data) have only been focused on single strains, a reduced set of phylogenetic groups or compared to distant mycobacteria[11,19,21]. Therefore, transcriptomic studies using sequence-based technologies and that take into account the whole MTBC genomic diversity are lacking. In this work, we have used RNA-seq data to study the transcriptomic signatures of different MTBC members, and identified differentially expressed genes using a novel phylogeny-based approach. In addition, we have revisited the mutational biases observed in MTBC populations and quantified the direct impact of individual genetic changes on transcriptional patterns. We have extended our analyses to include the impact of individual variants in methylation patterns and assessed their role in the regulation of in-vitro gene expression. Thus we provide quantitative evidence for the hypothesis[19,21], that the universal genome-wide mutational bias on MTBC leads to phenotypic plasticity at the transcriptome level.

## Results

**Samples, culture and DNA/RNA extraction.** We selected 19 strains from clinical samples which are representative of the MTBC global diversity (Supplementary Fig. 1). Each bacterial lineage (L1-6) is represented by at least 3 strains. Two replicates per strain were grown in standard 7H9 medium with the addition of 30 mM pyruvate to account for strains with potential pyruvate kinase mutations (Supplementary Data 1). Cells were harvested for DNA and RNA extraction at an $OD_{600}$ between 0.5 and 0.7.

The RNA extracted was sequenced on an Illumina HiSeq 2500 platform, and analysed using a custom analysis pipeline (see Methods for details). From the DNA extracted, long-read sequencing was performed on the PacBio RSII platform. In addition to transcriptome and long-read sequencing, short-read sequences for the selected strains were obtained from a previous publication[22] (see Data availability statement, Supplementary Data 2).

**Global transcriptomic patterns.** As a control, we first checked the agreement between sample replicates. We calculated the pairwise Pearson correlation between each pair of replicates. An almost perfect correlation (range 0.9996–0.9999) was achieved between each pair of replicates derived from the same strain. For subsequent analyses, the coverage data from the two biological replicates of each sample was merged.

Next, we surveyed the transcriptomic profile of the whole MTBC. A principal component analysis (PCA) was performed with the gene expression profiles of all the samples. Samples belonging to the same phylogenetic lineage grouped closely in the PCA (Fig. 1a). *M. africanum* (MAF, L5 and L6) and *M. tuberculosis* (L1-4) samples split along the first component (31% of the variance), thereby grouping according to their phylogenetic clade. Strains belonging to L1 were found between the MAF group and the modern lineages (L2, L3 and L4). As a further step, we performed an unsupervised hierarchical clustering (Euclidean distance, clustering method complete). The results agreed with the observations derived from the PCA, with the samples clustering according to their phylogenetic relationships (Fig. 1b). Furthermore, the intra-lineage genome-wide expression distance between samples is lower than the inter-lineage distances (Supplementary Fig. 2), supporting the idea that samples from the same lineage have a profile more similar to each other than with samples from other lineages. However, there were two exceptions. N0031 is part of L2 but its transcriptomic signature was different from other L2 strains. It has been previously reported that N0031 belongs to a rare, basal branch of L2 with a different transcriptomic profile than the more common, globally distributed L2 Beijing strains. The main difference between those clades is the overexpression of the *dosR* regulon in the L2 Beijing strains with respect to the rest of the complex including N0031[19]. On the other hand, N1177 belongs to L6 but it clustered with L5 samples. After the initial analysis, we realised that N1177 harbours a mutation in the *rpoB* gene (D435Y) that confers resistance to rifampicin. As mutations affecting the RNA-polymerase could have pleiotropic effects[23–26] it is not surprising that N1177 does not cluster together with the other L6 strains. Therefore, for subsequent analyses we removed N1177 as it may not be representative of the common L6 transcriptional profile.

As the RNA-seq profiles were congruent with the topology of the MTBC phylogeny, we investigated whether the number of differentially expressed genes between different clades was related to the genetic distance between them. We performed a Phylogenetically aware Differentially Expressed Genes (PDEG) analysis (see Methods for details) to infer the number of differentially expressed genes on each of the main branches of the phylogeny (Supplementary Data 3 and Supplementary Data 4). The results were highly variable, with a maximum of 42 PDEG genes in the branch leading to L6 and a minimum of 7 in the common branch of the modern lineages (Fig. 2a). We observed a reasonable trend in the data with the number of PDEG genes varying accordingly to the genetic distance between groups (Pearson's correlation value 0.57, *p*-value = 0.04, Fig. 2b). This suggests that the differences in the transcriptomic profiles between each group were accumulated

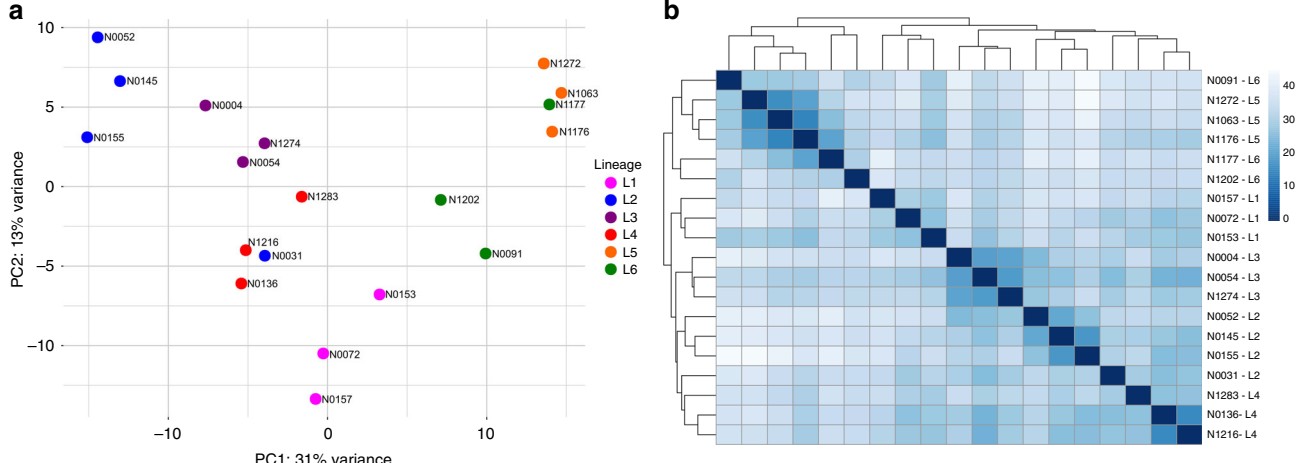

**Fig. 1** Global transcriptomic profiles of the samples. **a** The PCA plot shows that samples belonging to the same phylogenetic clade tend to group closely, except for two cases. **b** A cluster analysis reinforces the trend derived from the PCA, with almost all the samples belonging to the same lineage clustering together. The heatmap colour scale reflects the Euclidean distance between each sample, calculated from the complete transcriptomic signatures

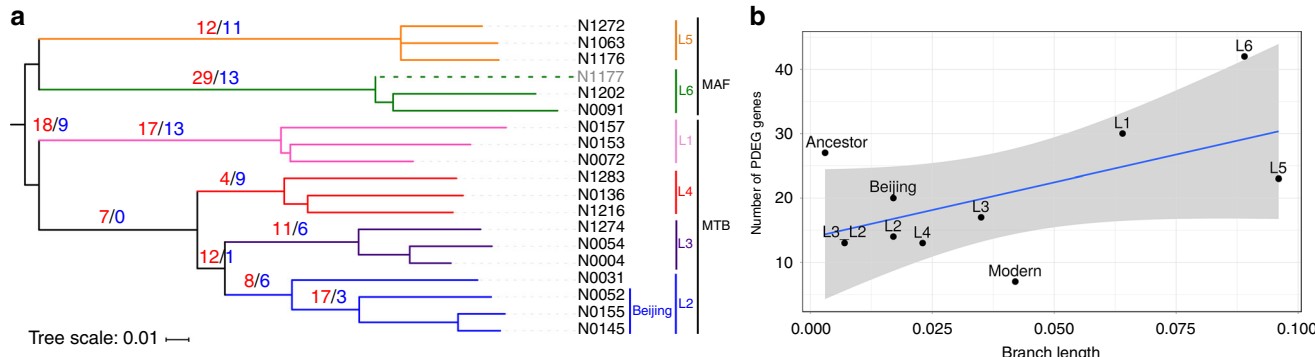

**Fig. 2** Gene expression changes across the MTBC phylogeny. **a** Number of genes differentially expressed (red up, blue down) in each of the main branches of the MTBC phylogeny. The phylogeny was constructed using Illumina sequencing data, the Maximum-Likelihood algorithm and a bootstrapping of 1000 replicates. Sample N1177 is included to shown the complete phylogenetic picture, but it was not included for further analyses. **b** Number of PDEG genes in each of the main MTBC branches plotted against the genetic distances

gradually as the MTBC lineages diverged. However, there were two branches that break slightly away from this trend. The split between *M. tuberculosis* and the two *M. africanum* lineages was defined by a short genetic distance but a high number of PDEG genes. In contrast, in the branch leading to the modern lineages we found the opposite situation. A complete list of the PDEG genes detected in each of the main branches can be found in Supplementary Data 4.

**Differential expression between phylogenetic clades**. We performed an enrichment analysis of Gene Ontology functions for the up-PDEG and down-PDEG genes for each of the branches analysed above. This analysis highlights the relative abundance of specific biological functions in a set of genes in comparison to the rest of the genome. Diverse biological functions appeared as upregulated and downregulated in each of the branches (Supplementary Data 5). Strikingly, most of them are related to host-pathogen interactions and key virulence metabolic processes. For example, the deepest phylogenetic split in the MTBC phylogeny is between MAF and MTB (Fig. 2a). 18 genes were significantly upregulated and 9 were significantly downregulated between both groups (BH adjusted *p*-value < 0.05, fold-change >1.5). Almost all of the *mbt* operon genes are upregulated in the MTB clade (*mbtI*, *mbtC*, *mbtH*, *mbtE*, *mbtG*, *mbtD*, *mbtB* and *mbtF*). These genes

code for the siderophore (mycobactin) system that is necessary for iron acquisition in iron-limited environments (i.e., macrophages)[27]. Genes *ctpG* and *ctpC*, that are involved in metal cation transport[28] also showed increased expression. Even though the *mbtJ* gene was not upregulated, its antisense transcript was highly overexpressed in MTB suggesting a differential regulation between MAF and MTB.

Although the MAF lineages 5 and 6 are geographically and genetically related, different studies have shown that there are phenotypic and genetic differences between both clades[29,30], and there is a high genetic distance between both lineages (Fig. 2a). Consequently, many PDEG genes appeared on deep branches that lead to extant strains in both groups. Toxin-antitoxin systems have been proposed to play a role in response to stress. Specifically, VapBC3 and VapBC5 are upregulated in the presence of moderately low pH conditions (i.e., the phagosome)[31]. In L6, we found that both these systems were upregulated. We also found upregulated genes related to the copper ion response (*lpqS*, Rv0967, Rv2642 and Rv2963) in this lineage. On the other hand, a number of important genes involved in parasitic functions such as virulence, persistence and macrophage infection were downregulated in L5 (Supplementary Data 5).

L1 is in between the so-called modern lineages (L2-L4) and the phylogenetically basal *Mycobacterium africanum* lineages

(L5-L6). One of the most upregulated genes in the L1 clade is *virS*, which encodes a transcriptional regulator essential for the transcription of the virulence-related *mymA* operon under acidic conditions[32].

Regarding each of the single modern lineages, the L4 branch had only 4 genes upregulated. Amongst them, Rv2159 and Rv2160A form part of an operon previously identified as being overexpressed in the *M. tuberculosis* H37Rv strain compared to *M. bovis* strains due to the loss of a transcriptional repressor[33]. In contrast, some of the genes involved in molybdopterin cofactor biosynthesis (*moaC* and *moaX*)[34] were downregulated in this branch, as well as part of the genes of the virulence-related *mce2* operon (*mce2C*, *mce2D*, *lprL* and *mce2F*).

In the branch leading to L3, 11 genes were upregulated while 6 were repressed. Surprisingly, the most upregulated gene in L3 strains was *oxyR*. This gene is involved in detoxification of ROS, contributing to the survival of the bacterium in the host, and also related to isoniazid resistance[35,36]. It has been previously reported that *oxyR* is inactivated in H37Rv, BCG, *M. africanum* and *M. microtti* due to several deletions that affect its translation[35]. Intriguingly, we have found that in L3, this gene had a 3-fold increase in expression compared to L2. The *ahpC* and *ahpD* loci upstream of *oxyR* were also overexpressed in L3 strains.

For L2, we have studied 4 representative strains. N0031 which belongs to a basal branch of this lineage and N0052, N0145 and N0155 which belong to the Beijing clade. As we noted in the global transcriptomic analysis, the N0031 transcriptomic profile was markedly different to those of the Beijing group. The DosR/DosS system was overexpressed in Beijing strains as previously reported[19], as well as the genes regulated by them. The DosR regulon is related to virulence and response to hypoxia[37]. In contrast, *plcD*, a gene related with extrathoracic progression of the disease and pathogenesis[38], was strongly repressed (log2 fold-change = −7.7).

**Mutation and selection lead to transcriptional plasticity.** Next we aimed to link specific transcriptional changes to genetic variants and to underlying mutational biases. It has been previously reported that mutations can create new Pribnow boxes (TANNNT motifs) that are recognised by sigma factor A, SigA, and lead to the overexpression of downstream genes[19,21,39]. To test the influence of such mutations, we scanned all the single nucleotide variants across the 19 samples that either create or disrupt TANNNT motifs. We found 603 variants that created new Pribnow boxes in at least one strain and 81 that disrupted

existing boxes (Supplementary Data 6). We investigated whether the observed impact on the Pribnow boxes resulted from stochastic mechanisms (i.e., genetic drift) or from non-random processes (i.e., selection). By comparing the number of expected versus observed occurrences (see Methods), we have obtained a probability of 0.006 for the observed number of disrupted boxes by chance and a probability of 2.5E−54 for the observed number of new boxes by random processes. So, it seems that non-random processes are acting to modulate the number of Pribnow boxes in the MTBC.

To test if selection is behind the observed number of boxes, we randomly introduced all the genomic mutations observed in the 19 strains and repeated the process 1000 times (Fig. 3a). We obtained a probability of 0 ($z$-score = 15.59) for having at least the same number of observed new boxes ($n = 683$) and a probability of 0.015 ($z$-score = 2.24) of having at least the same number of disrupted boxes ($n = 81$). Hence, it is unlikely that stochastic processes have been responsible for the observed occurrence of Pribnow boxes across the MTBC phylogeny. In addition, when we repeated this permutation test for other sigma factors' −10 consensus sequences such as SigE (cGTT), SigG (CGANCA) and SigJ (CGTCCT)[40], we observe the opposite pattern (Supplementary Fig. 3, Supplementary Notes). Our observations support the hypothesis that new SigA boxes are maintained by selection and not genetic drift.

We also noted that there was a remarkable difference between the number of new versus disrupted Pribnow boxes (ratio = 7.88). To get insights into the mechanism behind this figure, we randomly reordered all the mutations observed in our dataset by maintaining the alternative alleles but reshuffling the genomic positions. After that, we searched for new/disrupted Pribnow boxes in these 'reordered' mutations. A Fisher-exact test showed that there was no difference between real and reordered mutations in terms of new/disrupted boxes ratio ($p$-value = 0.39). Thus, the higher ratio observed between both type[41] of events is independent of the genomic context in which the new allele appears. It seems that these differences are caused by the type of substitution (TA alternative alleles could create TANNNT motifs, while mutation of wild-type TA bases disrupts them). It is known that there is a bias towards TA substitutions in bacteria, even in the case of GC-rich genomes such as the MTBC case[20]. Hence, this could be the cause of the notable difference between new acquisition and loss of TANNNT motifs. We next used a global dataset of MTBC clinical samples ($n = 4595$)[41], to check the alternative alleles derived from single nucleotide mutations and we observed that this pattern was also present across the

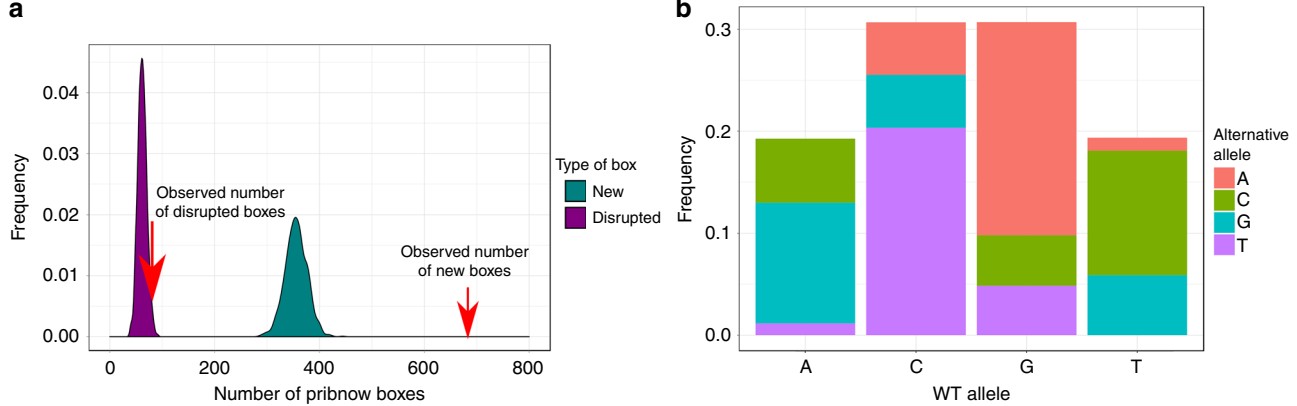

**Fig. 3** Non-random processes impact the emergence and disruption of Pribnow boxes. **a** Distribution of new (green) and disrupted (purple) Pribnow boxes in 1000 random simulations. Red arrows mark the observed value for each type of event in our dataset. **b** Mutation bias towards new A and T alleles inferred from 235,212 substitution obtained from 4595 clinical samples of the MTBC and normalised by GC content as in ref. [20]

MTBC (Fig. 3b). Thus, the mutational signature of the MTBC facilitates the appearance of new Pribnow boxes which, ultimately, supplies the bacteria with a higher transcriptional plasticity.

Finally, we looked for the potential impact on gene expression of these new and disrupted Pribnow boxes (Supplementary Data 6). We took into account only those mutations affecting the clades defined previously in the PDEG as the analysis of individual strains could lead to inconsistent results due to the lack of statistical power. First, new Pribnow boxes are over-represented among upregulated PDEG genes (chi-squared test, $p$-value = 2.78E−09). Second, when taking into account all genes, not just PDEG, we always observed higher expression of genes with a new Pribnow box due to a mutation compared to the closest relatives without the mutation. Conversely genes losing the Pribnow box because of a mutation have lower expression (Fig. 4a, wilcoxon test, $p$-value = 5.37−E09). A clear example is the observed overexpression of *oxyR* in L3 strains, potentially linked to a mutation (G2726105A) that creates a new Pribnow box (Fig. 3c).

Interestingly, 57 genes identified above as PDEG seem to be differentially expressed by means of a new or disrupted Pribnow box (Supplementary Data 6), meaning that ~26% of the transcriptional variability found across the MTBC main clades could be linked to single point mutations. New boxes were able to induce transcription in the sense or antisense direction, depending on the strand in which the mutation appeared (Fig. 4b), creating in some cases a complex regulatory scenario (see Supplementary Notes, Fig. 4c and Supplementary Fig. 4 for examples). To corroborate our results, data for L1 and L2 strains along with H37Rv grown in a different laboratory conditions were obtained from previously published work[19]. The same pipeline explained in the Methods section was applied to this new dataset. We see the same expression trends in those genes in which we originally linked a higher transcription rate to a lineage mutation generating a new Pribnow box (Supplementary Fig. 5).

**Differential methylation patterns across the MTBC.** From our RNA-seq analysis it is clear that there are marked differences in

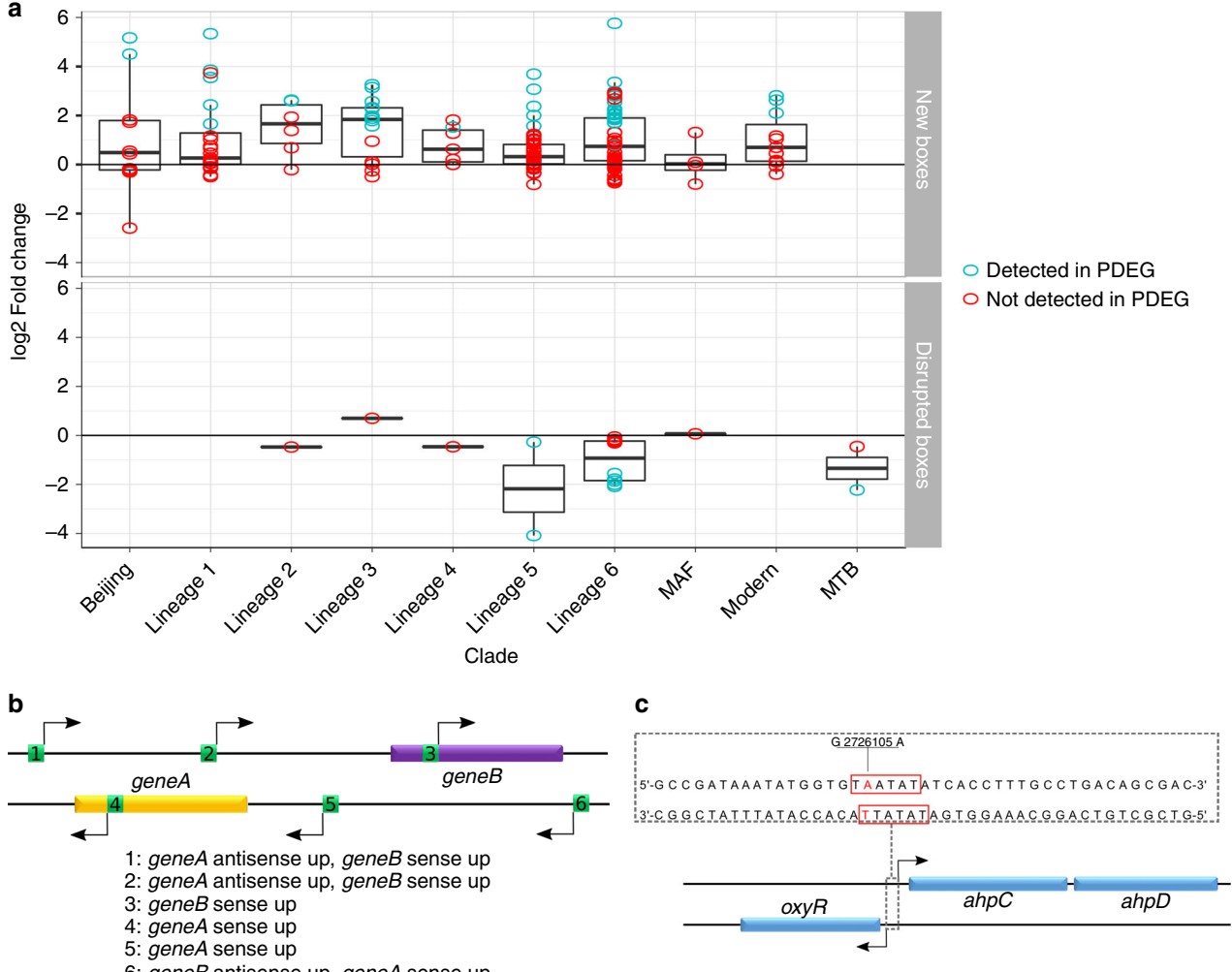

**Fig. 4** Impact of natural mutations in the appearance and disruption of Pribnow boxes. **a** Effect of the new/disrupted Pribnow boxes over the expression of downstream genes. New boxes tend to upregulate gene expression while disrupted boxes tend to downregulate transcription (wilcoxon test, $p$-value = 5.37-E09). Blue circles represent those changes in expression detected in the PDEG analysis (adj-pval < 0.05, log2 fold-change in expression > 1.5). Red circles represent subtle changes in gene expression, thus not identified by the PDEG analyses. **b** New Pribnow boxes can increase sense and/or antisense expression, depending on the genomic context in which the mutation appears. **c** The G2726105A mutation, common to all L3 strains, creates two new Pribnow boxes in the intergenic region of *oxyR* and *ahpC*. These new boxes are the potential explanation for the observed upregulation of *oxyR*, *ahpC* and *ahpD* in the L3 strains

gene expression between the main MTBC groups. Several mechanisms are known to impact gene expression in addition to sequence changes. Recent studies have shown that DNA methylation can have an effect on gene expression in bacteria[42]. To test the potential transcriptional effect of methylation in the MTBC, we sought to link differential methylation (DM) patterns between samples in our dataset with differences in gene expression. To do this, each sample was sequenced using the PacBio technology and analysed with the SMRT Analysis Software to identify methyltransferase recognition motifs (see Methods for details). Consistent with previous reports, we identified three main methylated motifs (CTCCAG, GATNNNNRTAC and CACGCAG) in almost all the samples[43–45]. The frequency of methylated sequences among these motifs was near 100% in almost all the samples. However, in some of the strains, the motifs were not methylated (frequency of methylated motifs 0%), suggesting that the methyltransferase that recognises this pattern is inactive (Supplementary Data 7).

These motifs have been previously reported to be recognised by three main MTBC methyltransferases MamA (Rv3263), HsdM/HsdS.1/HsdS (Rv2756c/Rv2755/Rv2761) and MamB (Rv2024c)[43–45].Interestingly, in two cases (N0052 and N0136) we observed that only a fraction of the motifs recognised by MamA were methylated (20% in N0052 and 56% in N0136). The sequences recognised by MamB and HsdM/HsdS.1/HsdS in N0052 and N0091, respectively, were also partially methylated along the genome (~70% of the sequences), suggesting that the activity of those methyltransferases is reduced, but not eliminated.

We wanted to identify the genetic variants that could be responsible for these functional differences. We therefore analysed the methyltransferase coding genes in the strains lacking methylation of one or more of the three motifs. This analysis resulted in the identification of several non-synonymous SNPs that could potentially be involved in the methyltransferase inactivation (or partial inactivation) (Table S7). Some of these variants have already been reported[44] while others are novel. Intriguingly, mamA in N0052 carries the same mutation as mamA in N0145 and N0155, although the activity in N0052 was only partially lost, compared to full loss of activity in the latter two, suggesting other genetic variants outside the gene may be having an effect. We expanded our analysis to the bigger dataset reported above ($n = 4595$) to get a global picture of the methyltransferase conservation degree. Some of these variants located deep in the phylogeny affected complete lineages while others were more recent and affected only a subset of strains (Fig. 5a). For example, of the W136R and G152S mutations that were found in the MamA inactive strains, G152S was found in a subset of L4.3.3 strains and a small clade of L1.1.2 (i.e., it is homoplastic) while W136R affected a subset of L1.2.1 samples. Interestingly, a new mamB variant (D59G) was also found in these strains potentially linked to MamB inactivation. On the other hand, a T393A variant was found to affect hsdM in a subset of lineage 6 strains potentially affecting their methyltransferase activity.

To gain a wider perspective on the main MTBC methyltransferase diversity, we analysed all the variants present in these genes in the larger dataset (Supplementary Data 8). The dN/dS values for mamA (0.75) and mamB (0.76) were slightly higher than the mean dN/dS value for non-essential genes[46] (0.66). In contrast, HsdM/HsdS.1/HsdS show a lower accumulation of non-synonymous mutations, with the gene that encode for the specificity unit hsdS (0.76) having a similar value than mamA and mamB, and the genes that encode for the methyltransferase unit hsdM (0.5) and the specificity unit hsdS.1 (0.58) showing a value similar to that of the essential genes[46] (0.53). Despite gene-wide conservation of the methyltransferases we observed the accumulation of functional mutations in the form of new stop codons.

For example hsdM accumulates 5 stop codons in different parts of the phylogeny suggesting that either the gene is under weak selection (contradicting the low dN/dS observed) or that specific mutations on the gene have been selected during evolution even though we do not observe any impact on expression profiles of unmethylated strains.

**DM impact on transcription is subtle and lineage independent**. DM in regulatory regions has been reported as potentially affecting gene expression in H37Rv[13]. We wanted to check if DM naturally present in our strains could be linked to differential gene expression. To achieve this, we looked for SigA recognition motifs (TANNNT / GNNANNNT[21]) in gene promoter regions (−50 bp upstream the TSS previously defined[39]) that overlap with methyltransferase recognition motifs. We managed to identify SigA recognition motifs for 13 genes overlapping with the MamA motif, 22 with the HsdM/HsdS.1/HsdS motif and 2 with the MamB motif (Fig. 5c). To account for differential gene expression due to DM and not for other evolutionary reasons, we compared gene expression values in strains which belonged to the same lineage but in which the specific methylase was either active or inactive. This was the case for MamA in L1 and L2, HsdM/HsdS.1/HsdS in L4 and L6, and MamB in L4 (Table S7).

First, we compared the expression of the 13 genes identified in both situations (MamA activated or inactivated) in L1 and L2 strains. We observed that almost all genes increase their expression values in the methylated strains in both lineages, matching previous observations in H37Rv[13] (Fig. 5b). However, we identified some exceptions in which the gene expression behaved differently in each lineage. For example, Rv3727 showed a lower expression in methylated strains (N0072 and N0153) than in the non-methylated strain (N0157) of L1. Rv3727 is regulated by the transcription factor Rv0022c which had an early-stop codon in the N0157 strains[17]. This type of polymorphism could be the cause of the discordant results. For MamB, we only found 2 genes where SigA and MamB motifs overlapped. Even so, for these two genes we observed the same effect as in MamA DM strains.

However, for HsdM/HsdS.1/HsdS we did not observe this pattern of changes in gene expression. The overlap between SigA recognition motifs and HsdM/HsdS.1/HsdS motifs in the regulatory regions seemed to have no impact on gene expression. In some cases the genes increased their expression in the non-methylated strains while some others behave in the opposite manner. Moreover, this behaviour was not congruent in L4 and L6 as half of the genes showed the same regulatory response in both lineages while the other half behaved differently in each lineage. In summary, MamB and MamA methylation over SigA motifs seems to cause a similar effect independently of the strain genetic background while HsdM/HsdS.1/HsdS seems to have no effect.

In addition, we searched for other sigma factors that could potentially have an overlap between their recognition motifs and methyltransferase recognition motifs. We found that the SigB recognition motif (NNGNNG)[47] could overlap and we applied the same analysis as for TANNNT. However, DM seemed to have no effect on promoter regions having NNGNNG motifs. It is known that SigB plays a role during stress response[48] but it is dispensable for growth. As the RNA-seq samples were collected during exponential growth (applying no stress), DM over SigB influenced genes could show little or no differential expression.

**A different mechanism for HsdM gene expression regulation**. While minor effects on gene expression were found linked to MamA and MamB (Fig. 5b) we were surprised that no effect

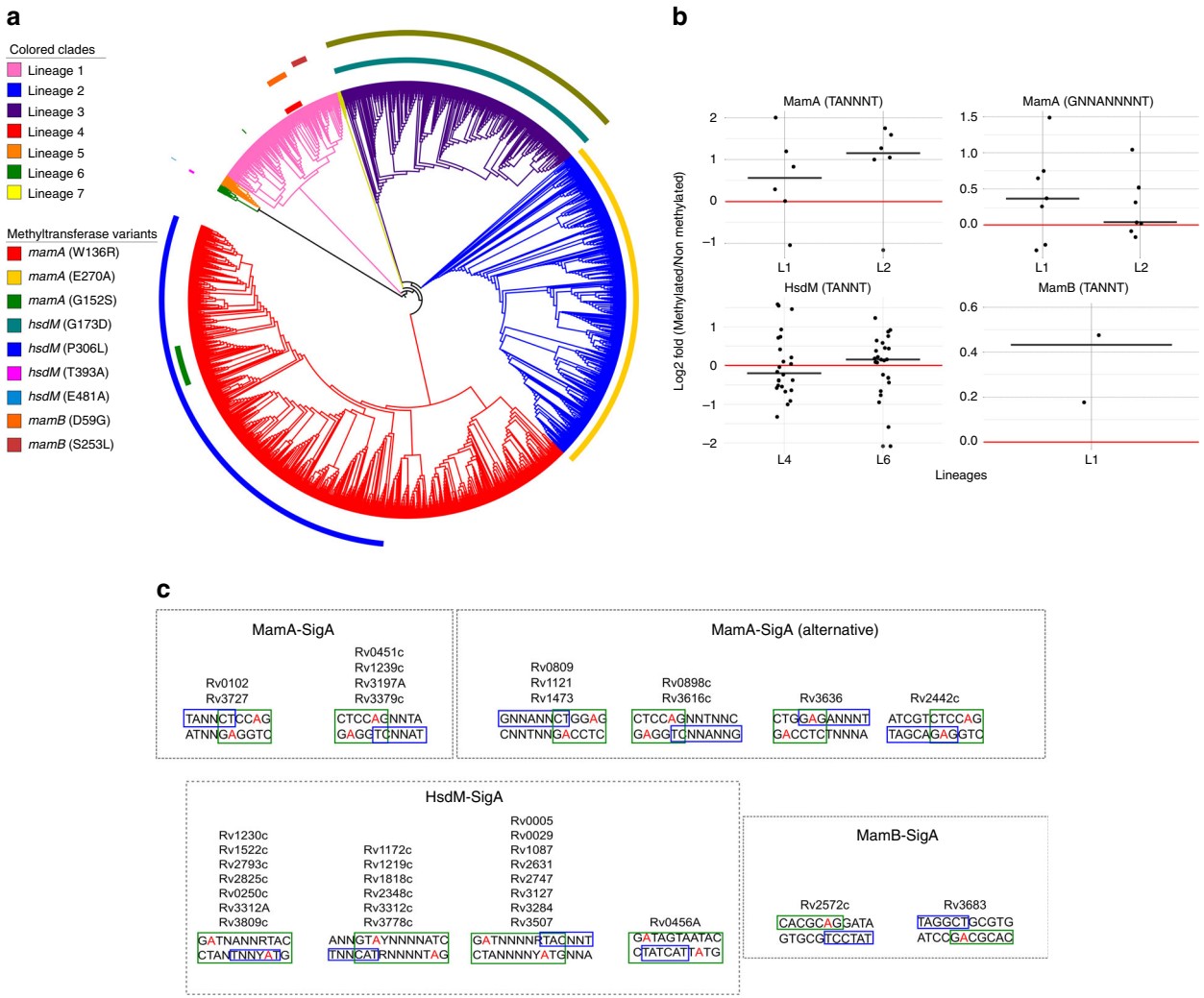

**Fig. 5** Methyltransferase activity of the MTBC. **a** Distribution of the characterised mutations that potentially impair methyltransferase function on a global dataset (n = 4595 samples). **b** Gene expression differences between SigA recognition motifs differentially methylated by each of the three methyltransferases. Red line marks a 0 fold-change in gene expression (no differences). The expression of each gene was tested in both situations, methylated and non-methylated strains, in independent lineages (when possible). **c** Different overlapping patterns found between SigA recognition motifs and the methylated motifs. The red adenines in the motif are the methylated ones

could be linked to HsdM, particularly as HsdM is the one that has accumulated more stop codons during the evolution of the MTBC (Supplementary Data 8). This suggests that it is either under weak selection, randomly accumulating inactivating mutations, or that it has a functional role and specific mutations in HsdM have been selected in different parts of the MTBC phylogeny. To discriminate between these two possibilities we mimicked a stop codon mutant for HsdM by deleting the *hsdM* gene in a N1283 background (L4). Compared to the other two L4 strains in the transcriptome dataset HsdM is fully functional in N1283 (Table S7) which allow us to compare it to transcriptomic profiles of unmethylated MTBC strains.

We then performed a transcriptomic analysis comparing the *ΔhsdM* strain and the wild-type. An initial analysis showed differences between the strains, as the transcriptomes split into two groups in a PCA analysis (Fig. 6a). In the DE analysis, we observed that these differences were mainly driven by a small number of genes (BH adj-pvalue < 0.05 and log2 fold-change >1, Supplementary Data 9, Fig. 6b). In N1283-*ΔhsdM*, several genes were increased in expression in comparison with the wild-type. First, *hsdS.1* expression was increased in the mutant, suggesting

that its regulation is linked to *hsdM* (which is found upstream in the H37Rv genomic context). In addition, a set of 7 consecutive genes (Rv0081−Rv0087), potentially forming an operon, were found to have increased expression. Interestingly, Rv0081 is a transcriptional hub involved in the regulation of multiple genes[12,17,49], including the hyc-family genes, which have homology to the so-called EHR (energy-converting hydrogenases related) complexes. Evolutionarily, EHR proteins stand between complex 1 and NiFe-hydrogenases[50] and their functions have yet to be determined. The EHR complex of the MTBC is with high certainty not a functional hydrogenase, because *M. tuberculosis* lacks the cluster of assembly genes needed to mature NiFe centres and insert them into the protein[51]. In contrast, Rv1813c, Rv0080, Rv3131 (all hypothetical proteins) were decreased in expression in the mutant, as well as *ctpJ*. Thus, HsdM methylation has an effect on gene expression, but the mechanism seems to be different to that of MamA and MamB, as the genes reported above did not have any overlap between SigA and HsdM motifs. Moreover, we found no bases methylated by HsdM near these genes, suggesting an indirect effect of HsdM DM on gene expression.

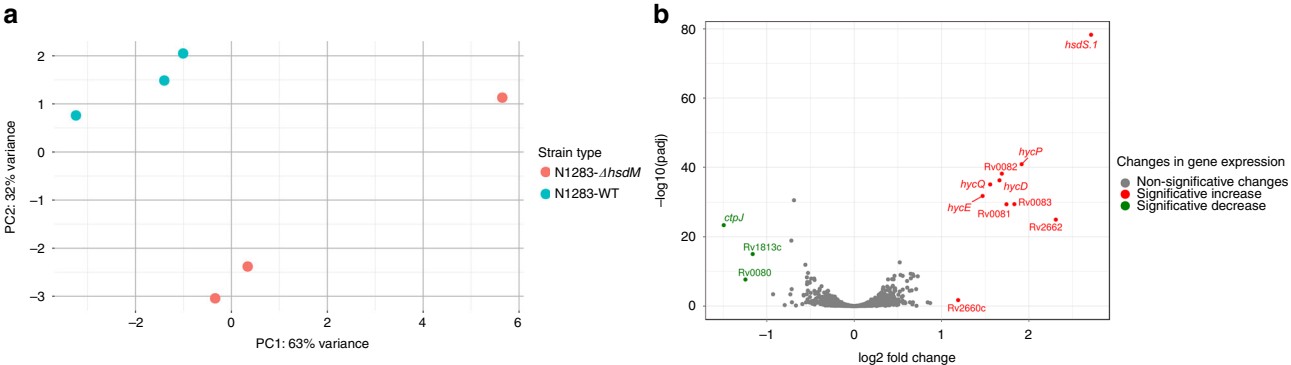

**Fig. 6** Gene expression differences due to an *hsdM* deletion. **a** Overall transcriptomic profiles of the wild-type versus the *ΔhsdM* strains. **b** Volcano-plot of the gene expression differences of the wild-type versus de mutant strains. A small numbers of genes showed differential expression (3 downregulated and 10 upregulated)

## Discussion

Our results show that the different MTBC clades have their own transcriptomic signature. Each main lineage is defined by a transcriptomic landscape, that clearly separates it from the rest of the lineages. We have shown before that transcriptional regulators are not conserved across lineages[17]. Now, we show that even single point mutations may totally change the transcriptional profile of a strain. An example is strain N1177, which carries a single mutation in the *rpoB* gene conferring rifampicin resistance which modified the transcriptional levels of multiple genes. Likewise, a single mutation generating a new SigA recognition motif increases the expression of the DosR regulon in the three Lineage 2 Beijing strains but not in the basal Lineage 2 strain (Fig. 1a,[19]). Our phylogeny-based approach has allowed us to identify gene expression changes that took place during the evolution of the MTBC. We have observed that, as the MTBC diverged into the different lineages, expression of key host-pathogen and metabolic genes also did so. This provides further evidence that lineages of MTBC likely reflect adaptation to different human populations.

Modification of gene expression could be a rapid mechanism for the physiological adaptation to a new environment without the need to substantially change the genome. This could have been the case when MAF and MTB split from a common ancestor, with a relatively short genetic distance, but many genes changing their expression. We propose that a sudden environmental change (possibly a change in host population) rapidly selected nascent phylogenetic groups that behaved differently in terms of gene expression, or that standing variation in regulation allowed the ancestor to differentially specialise in different environments. In accordance with this, the enrichment in genes involved in metal homoeostasis may be related to different concentrations of ions in different host populations or animals[52].

The analyses of the genetic bases of expression differences between phylogenetic clades reveals an interplay of natural selection and mutational processes. We show that at least 26% of the core expression differences between lineages were due to single point mutations creating new Pribnow boxes in gene regulatory regions. This number may be higher as we have not analysed indirect regulatory effects. The number of new Pribnow boxes is more than expected by chance and thus selection likely played a role in fixing expression differences. Importantly, the underlying AT mutational bias across the genome has been a source of expression diversity through random generation of new Pribnow boxes, as previously theorised[21]. The reason why selection is apparent for SigA motifs but not for other sigma factors

remains unclear but at least two non-mutually exclusive explanations are possible. On the one hand, SNPs impacting SigA recognition motifs can have an impact across environmental conditions while SNPs for other sigma factors only will be relevant for specific conditions. They may happen but are more difficult to detect in our analyses. On the other hand, SigA motifs are enriched in AT bases and thus it is not surprising that new SigA motifs are generated at a faster pace leaving more room for selection to act. However, the fact that we observe less boxes than expected by chance in most non-SigA sigma factors suggests that negative selection is acting on some of them.

It seems clear from our results that there has been a convergence of the methylation patterns in the different phylogenetic groups of the MTBC, instead of a lineage-specific pattern as proposed previously[44]. Equivalent phenotypes (non-methylation of specific motifs) appear to be produced by different genetic variations (Supplementary text). For example, W136R mutations in a subset of L1 strains seem to have the same effect as E270A in a subset of L2 strains, impairing MamA activity. Our results also show that methylation seems to play a minimal role in shaping in-vitro gene expression. We have not been able to detect a regulatory impact for the main methyltransferases, except for a subtle effect on few genes having overlapping SigA and MamA/MamB recognition motifs, consistent with previous reports[13]. This could be due to our inability to identify genes that are actually influenced by the methyltransferases, as the *Δhsdm* strain shows differential expression in genes that we had not previously identified as potentially influenced by HsdM. MamA/MamB methylation motifs do overlap with SigA recognition motifs, affecting the transcription mediated by SigA, however, this seems to not be the case for HsdM.

In summary, we have carried out a comprehensive comparison of transcriptomes and DNA-methylomes of nineteen clinical isolates representative of the global phylogenetic spectrum of the human-adapted strains of the *Mycobacterium tuberculosis* complex. Patterns of differential transcription between lineages reflected constitutive expression of genes that are normally regulated in response to environmental cues, as a result of mutations that introduce novel TANNNT Pribnow boxes and mutations that impair the function of transcriptional repressors. The role of methylation is more elusive but it is clear from the pattern of inactivating mutations that methylases are not conserved across the MTBC. Isolated from the opportunity to generate diversity by horizontal gene transfer[41], transcriptional adaptation may allow *M. tuberculosis* isolates to optimise their infectivity and transmission in subtly differing environments provided by different human host populations.

## Methods

**Culture conditions**. All cultures were grown in ink wells containing 10 ml Middlebrook 7H9 OADC medium supplemented with 30 mM sodium pyruvate to account for pyruvate kinase mutations in L5 and L6 (Supplementary Data 1). Cultures were grown on orbital shakers at 80 rpms at 37 °C. For each strain, two biological replicates were cultivated.

**RNA isolation and Illumina sequencing**. For RNA extraction cultures were grown to $OD_{600}$ of 0.5–0.7. Ten millilitre aliquots were spun down and immediately processed with TRIZOL reagent according to manufacturer protocols. Cells were harvested from exponential cultures and RNA extracted using the Direct-zol™ RNA Kit from Zymo according to manufacturer's instructions. From the RNA extracted, ribosomal RNA was depleted by using a Ribo-Zero Magnetic Kit. After that, sequencing libraries were prepared using the TrueSeq stranded Illumina protocol and sequenced on an Illumina HiSeq 2500 platform.

**DNA isolation**. For DNA extraction cultures were harvest between $OD_{600}$ of 0.5–0.7 by spinning down 5 ml culture and immediately starting DNA extraction by CTAB method[53].

**RNA-seq pipeline**. Fastq files qualities were assessed using FastQC[54]. Trimmomatic, a programme that uses a dynamic trimming approach[55], was used to remove bases from the start and the end of the reads when its quality was below 20. Reads were mapped to the H37Rv reference strain[56] using BWA-mem algorithm[57]. Potential duplicates were removed by using the MarkDuplicates option from the Picard tools package[58]. Bedtools[59] was used to calculate the read coverage for each genomic feature. To precisely report the coding and non-coding coverage, each read was classified according to the strand from which it was initially derived.

**Transcriptomic analysis**. The statistical analysis was performed using the R statistical language[60], specifically the DESeq2 package[61]. The input data was the count table containing the coverage information for each feature for all the samples. The PCA and the hierarchical clustering were performed by previously normalising the count data across samples and scaling it into a log2 scale, by using the rlog function from the DESeq2 package.

For the analysis of Phylogenetically aware Differentially Expressed Genes (PDEG), we performed a two-step process. First, we identified all the genes having differential gene expression (adjusted BH $p$-value < 0.05 and log2 fold-change >1.5) between each pair of phylogenetic groups with a common origin (for example L5 and L6, MAF and MTB, etc). Therefore, we identified the genes changing their expression between these groups. This information however, is not enough to assign the expression change to one group or the other. We cannot know if an increase in gene expression for one gene is due to an upregulation in one group or to a downregulation in the other. To resolve this, for each gene identified as differentially expressed, we compared its expression value in each of the two groups against the rest of the MTBC samples. This analysis allowed us to identify the group in which the change in gene expression took place and the direction of this change. Finally, we assigned all the changes in a group to the tree branch common to this clade. For this part of the analysis, sample N1177 (L6) was excluded, as the *rpoB* mutation alters its transcriptomic signature and it is therefore not representative of the L6 transcriptomic signature.

Genes with deletions in each of the groups were not taken into account in the pairwise comparisons, as they result in false positive signals. These genes were identified by mapping long-reads obtained from PacBio sequencing against the H37Rv reference genome, and assessing the genomic coverage (Supplementary Data 10). PE/PPE, phages and repetitive genes have not been taken into account in any of the analyses, as their sequenced reads are prone to map erroneously (Supplementary Data 11). The enrichment analysis in GO functions was performed using the BiNGO tool[62]. BiNGO identifies the most abundant functions in a subset of genes, compared to all functions present in a complete genome using a hypergeometric test (sampling without replacement).

**FASTQ mapping and variant calling from the Illumina data**. For each of the analysed strains, we downloaded the publicly available genomic data from a previous work[22] (Supplementary Data 2). We analysed the data using a published pipeline[63]. Briefly, fastq files were trimmed using fastp[64] and aligned to the MTBC most likely ancestral genome[46] using BWA-mem[57]. Potential duplicated reads were removed using Picard tools[58]. Samtools[65] and VarScan[66] were used to perform the variant calling. A SNP was called if it was supported by at least 20 reads, it was found in a frequency of at least 0.9 and was not found adjacent to an indel area or in areas of high accumulation of variants (defined as more than 3 variants in a 10 bp defined window). Variants were annotated using SnpEFF[67]. Variants found in phages, PE/PPE genes or repeated regions were filtered out as they are difficult to map and induce accumulate many false calls. All the variants found were used to generate a multiple alignment of variant positions with all the strains analysed. An MTBC phylogeny was calculated by using the RAxML programme[68] with the GTRCATI model of evolution and represented with the iTOL software[69].

**Creation and disruption of Pribnow boxes**. Using the MTBC ancestor genome as a template, we introduced all the mutations found in the dataset, and look for new/disappeared TANNNT motifs with the fuzznuc tool included in the EMBOSS programme[70]. By doing this, we have obtained the number of affected Pribnow boxes in our dataset. Three independent tests were performed to calculate the probability of the observed new/disrupted SigA recognition motifs by chance.

To calculate the probability of appearance or disruption of Pribnow boxes we have first scanned the MTBC ancestor genome looking for the 'ancestral' TANNNT motifs, or for motifs that could result in TANNNT motifs by introducing one single mutation (VANNNT, TANNNV, TBNNNT). In parallel, from the observed number of variants in the MTBC dataset, we calculated the probability of a non-A (B), non-T(V), A and T mutations. After that, we calculate the expected disruption of boxes by inferring the probability of non-A or non-T mutations to fall in the 1st, 2nd or 6th position of the 'ancestral' motifs. The expected generation of new boxes was calculated by inferring the probability of A and T mutations to fall in the corresponding VANNNT, TANNNV and TBNNNT motifs. In a last step, we have used a Poisson distribution to calculate the probability of the expected versus the observed number of disruptions in our dataset (Supplementary Fig. 6a).

A random permutation test was performed by keeping the alternative alleles for the 8093 SNPs found in the global dataset, but randomly assigning a new genomic position in which those SNPs appear. Later, we scanned for new/disappeared TANNNT motifs with the fuzznuc tool. This process was repeated 1000 times. With the number of new/disrupted boxes in this 1000 simulations we calculated a cumulative empirical distribution of expected Pribnow boxes affected by random mutations. Later, we compared these distributions with the number of boxes affected by the real variants (Supplementary Fig. 6b).

Finally, we have reshuffled the 8093 mutations so the alternative alleles were randomly assigned to genomic positions that initially harbour other variants. Again, we impacted the MTBC ancestor genome with these variants and assessed the number of new/disrupted Pribnow boxes by using fuzznuc. The results obtained were confronted against the number of new/disrupted boxes with the real 8093 variants in a chi-squared test (Supplementary Fig. 6c).

**Genomic DNA isolation for PacBio Sequencing**. DNA was prepared and sequenced on the Pacific Biosciences RSII machine as described in a previous work[71]. Briefly, we used template preparation kit version 3.0, polymerase binding P6 version 2, and sequencing reagents version 4.0 (C4). Data were captured using 3-h movies. Each sample replicate was sequenced on two to four chips to get enough genome coverage for the detection of the methylated patterns.

**HsdM mutant construction**. The gene *hsdM* was deleted in strain MTB N1283 by specialised transduction[72]. Transductants were recovered on 7H10 OADC plates containing hygromycin (75 μg/ml). Mutations were confirmed by three-primer PCR using primers HsdM_L, HsdM_R, and Universal_uptag, listed in Supplementary Data 12.

**Methylation analysis and variant search**. DNA isolated from the cultured samples were used for PacBio sequencing. The SMRT Analysis Software from PacBio[73] was used to detect methylation patterns in the PacBio sequencing data. Sequencing reads from both biological replicates per strain were merged to assess a higher sequencing depth. The Modification_and_Motif_Analysis protocol was used as defined in the SMRT manual. This protocol detects the Interpulse Duration (IPD) to classify one base as methylated. After that, it looks for over-represented methylation motifs in the genome.

For those strains lacking at least one of the three main methylated motifs present in the rest of the dataset, we looked for non-synonymous variants affecting the methyltransferases. These variants, that potentially affect the methyltransferase function, were also scanned in a dataset of 4,595 strains representative of the MTBC global diversity[41]. The potential effect of the non-synonymous variants over gene functionality was assessed by using the SIFT4G tool[74]. dN/dS values for the methyltransferases were calculated as described previously in[37]. Briefly, by using the observed non-synonymous and synonymous variants in these genes, and the potential synonymous and non-synonymous substitution sites for each gene (calculated using the SNAP tool[75] and without distance correction), the dN/dS for each gene was defined in Eq. 1:

$$\frac{\text{Non} - \text{synonymous variants} \setminus \text{Non} - \text{synonymous sites}}{\text{Synonymous variants} \setminus \text{Synonymous sites}} \tag{1}$$

**RNA-seq analysis linked to differential methylation**. We used fuzznuc from the EMBOSS package[70] to identify genes whose conserved −10 TANNNT motif overlapped with identified methylated motifs. The potential effect of methylation over the expression of these genes was assessed by comparing the expression values in strains with a similar genetic background (same lineage), but having differential methyltransferase activity.

**Reporting summary**. Further information on research design is available in the Nature Research Reporting Summary linked to this article.

## Data availability

All the new data generated for the present study was submitted to ENA under accession numbers PRJEB8783 (PacBio data) and PRJEB9763 (transcriptomic data). Whole-genome sequencing of N0153, N117 and N1063 was deposited under accession PRJEB31443, while the sequence data from the rest of the samples comes from another publication[22] and can be found under accession number PRJEB27802. The source data underlying Figs. 1a, b, 2b, 3a, b, 4a, 5b, 6a, b are provided as a Source Data file. All other relevant data is available upon request.

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

## Acknowledgements

This project has received funding from the European Research Council (ERC) under the European Union's Horizon 2020 research and innovation programmes 638553 (TB-ACCELERATE to I.C.) and 637730 (MtbTransReg to T.C.). M.B. was funded by the National Institute of Health grant AI119573. W.R.J. was funded by the National Institutes of Health grant AI26170. In addition, this work was funded by projects SAF2016-77346-R from Ministerio de Economía y Competitividad (Spanish Government) and AICO/2018/113 from Generalitat Valenciana (to I.C.), BFU2014-58656-R, BFU2017-89594-R from Ministerio de Economía y Competitividad (Spanish Government), PRO-METEO/2016/122 from Generalitat Valenciana (to FGC), Wellcome Trust grant 098051 to the Sanger Institute. A.C.O. is recipient of a FPU fellowship from Ministerio de Educación y Ciencia FPU13/00913 (Spanish Government).

## Author contributions

I.C. and T.C. conceived this work. M.B. performed all the experiments. A.C.O. analysed the data. C.B. constructed the sequencing libraries. A.C.O., M.B., S.G., J.P., T.C. and I.C. wrote the first version of the draft. A.C.O., M.B., S.G., J.P., T.C., I.C., F.G.C., D.Y., W.R.J. critically reviewed and contributed to the final version of the paper.

## Additional information

**Competing interests:** The authors declare no competing interests.

