## [Peer Review File · Nature Communications]

Reviewers' comments:

Reviewer #1 (Remarks to the Author):

The authors performed transcriptome, genome and methylome analysis to systematically screen the differentially expressed genes between MTBC clades, and further discovered that genomic variants and their disruption of methyltransferase have critical role on expression regulation. They also searched a lot of public datasets to gain a full perspective on their main conclusions. The findings obtained by the authors are informative to describe the relationship among genome, transcriptome and methylome from MTBC strains. However, as some bioinformatics papers, the manuscript is not well organized since the authors tried addressing too many results and message. The following concerns should be addressed.

1. The gene expression and methylation states are real-time, unstable, and they often varied with bacterial growth. The authors should performed the RNA and DNA extractions using the same batch of cells at the same growth timepoint/state. Otherwise, it is not reasonable to conduct comparative analyses among transcriptome, methylome and genome.
2. Line 123-128: the authors remove strain N1177 from the analysis based on the PCA result and an *rpoB* mutation. The author said that *rpoB* mutation (D435Y) affected the RNA-polymerase and transcription. Pls provide the related reference. In addition, *rpoB* mutation is a type of common one in TB strains, so the authors should search for the *rpoB* mutation in all the MTBC strains and remove them from the study.
3. Line 119-128: the explanation for preservation of stain N0031 in the study is inadequate. If the expression of stain N0031 is different from other L2 strains, the authors should also remove the strain like strain N1177.
4. Line 280-285: the authors described that 26% transcriptional variability found across the MTBC main clades was due to single point mutations. As one of the main conclusions in this study, I recommend that the authors validate some altered gene expressions by qPCR.
5. L365-384: the authors described that two methyltransferases, MamA and MamB could influence the expression of some genes through the overlap between the methyltransferase recognition motifs and SigA recognition motifs. However, this is not the case for HsdM. It is better if the authors could further describe the pattern of various HsdM recognition motif sequences (GATN₄RTAC), such as GATN₄ATAC and GATN₄GTAC.

Reviewer #2 (Remarks to the Author):

The manuscript by Chiner-Oms and colleagues describes a transcriptomics analysis of multiple strains of Mycobacterium tuberculosis from distinct evolutionary lineages. The authors present a substantial amount of data that includes a comprehensive transcriptomic analysis that encompasses 19 M. tuberculosis strains from six lineages, global methylation profiling of these strains to explore links between gene expression and differential methylation of promoter elements, and analysis of a methyltransferase mutant vs wild type. The authors present evidence that lineages of M. tuberculosis have shared transcriptomic signatures, and hypothesise that this transcriptional diversity links to phenotypic diversity and ultimate success of the lineages.

I have come comments I would like to see addressed.

A caveat of the study is that only a single in vitro growth condition was studied, namely exponential growth in laboratory media, and this is acknowledged (line 391). Looking through the detail of the manuscript I wasn't clear exactly at what point in the growth curve the strains were harvested. It

would be necessary to describe whether the growth rates of the strains were similar under the in vitro conditions used; otherwise it could be argued that the transcriptional differences across strains are simply a function of in vitro growth rate differences. For example, it is mentioned that pyruvate was added to media “to account for strains with potential pyruvate kinase mutations”; does this mean pyruvate was the sole carbon source used? Or was glucose or glycerol also present? Choice of carbon source by strains may affect growth kinetics and hence global transcriptional activity.

Line 119-128: The authors show that the N0031 strain does not match the transcriptional profile of other strains in the lineage because of a mutation in a master transcriptional regulator, *dosR*. Likewise, the N177 strain did not cluster with other L6 strains due to a mutation in *rpoS*, conferring rifampicin resistance. Hence the results from both of these isolates show that strains within a lineage can have a divergent transcriptional profile from other strains, and that this may particularly be the case when strains are drug-resistant. Perhaps this latter point, with relevance to drug resistance, could be more explicitly made in text.

As further evidence to support the authors' hypothesis, I wondered whether a global analysis of the lineages would reveal significantly more mutations in genes encoding transcriptional regulators? For example, it has been shown that mutations in the PhoPR system of *M. tuberculosis* can affect the expression of genes which may be involved in the distinct host preference of *M. tuberculosis* complex members. Similarly, serine-threonine protein kinases show differences across *M. tuberculosis* strains that impact on global gene regulation. Might such master regulators harbour greater genetic diversity than would be expected by chance, and hence play into transcriptional plasticity across lineages? I realise that the authors were focusing on 19 strains for transcriptome analyses, but augmenting this with a global analysis of available *M. tuberculosis* genome sequences would fit well within the manuscript and may provide further evidence that would support their hypothesis.

Line 195-201, and Fig4c: Lineage 3 strains were shown to have increased expression of *ahpC-ahpD-oxyR*, and that this may be linked to the creation of a new Pribnow box. It would have been good to have seen this experimentally verified by plasmid reporter constructs.

Line 229: “...we randomly introduced all the mutations observed in the genome...”. By ‘the genome’, do the authors mean mutations across all lineages, including *M. africanum*?

Line 398-400: “...the natural variability of HsdM found in the MTBC suggests that there may be some biological relevance associated with this protein.” What is meant by ‘natural variability’, and ‘biological relevance’ here? I think more precision would be needed here as to why *hsdM* was selected for mutant construction and transcriptome analyses.

Line 440-1: “This is another proof that lineages of MTBC likely reflect adaptation to different human populations”. It would be better to say “This provides further evidence that lineages of MTBC likely reflect adaptation to different human populations”.

Reviewer #3 (Remarks to the Author):

Clinical isolates of *Mycobacterium tuberculosis* (*Mtb*) have been repeatedly shown to have meaningful phenotypic heterogeneity in virulence, however our current understanding of the mechanisms driving these differences is incomplete. Here Chinar-Oms et al. utilize transcriptomics and whole-genome methylation analysis of a diverse set of clinical isolates to track the divergence of *Mtb* expression profiles across its evolutionary history. In the process, they present a framework for analyzing

transcriptome changes across the phylogenetic tree which allows them to look for mutations and methylation pattern alterations which correspond with altered gene expression.

Overall the manuscript provides a wealth of new data which was not previously available, however there are several areas where the authors would benefit from further clarifying their methods and supporting their conclusions with statistical tests.

In particular, the authors focus on ascribing genetic changes in pribnow boxes to alterations in gene expression, however their analysis linking these two phenomena is not supported by any statistical test. Clearly not all mutations in pribnow box sequences identified by sequence alone will mediate a change in gene expression, however if they want to ascribe these mutations to the altered expression then there must be some additional information provided. For example, examination of the RNA-seq data may help identify if there is actually increase transcription coinciding with the new pribnow boxes which could further bolster their case at genes where the expression appears affected.

The global analysis of methylation among these clinical isolates, and corresponding identification of mutations in methyltransferase machinery is also very interesting however the data presented on the hsdM mutant do not appreciably add to the paper.

Finally, the authors focus almost exclusively on differences among the strains at the level of somewhat arbitrarily designated lineages. To that point, the most divergent lineage 2 isolate in fact doesn't appear much like other lineage 2 isolates transcriptionally. Given the rich nature of this dataset, it would be illuminating to report what fraction of transcriptional variation can be accounted for by the lineage designation as opposed to differences among individual strains.

Abstract:

Line 32: "The alterations in gene expression" sentence is confusingly worded.

Introduction:

Line 42:

Monomorphic seems like a strange term to use. Perhaps clonal or non-recombining.

Results:

Line 107:

It is not clear to me why you would merge the data from two replicates rather than treat them as actual replicates in further analyses.

Line 110:

It isn't clear if the PCA was performed on data which is coverage normalized or whether it was just log₂ transformed. It should be.

Line 127:

The authors state sample N1177 is removed because rpoB mutant which is probably messing up all sorts of stuff. This isolate is still included in Figure 2a and it is unclear if it was incorporated into the calculations for Figure 2b.

Line 143:

Change contrarily into "In contrast"

Line 155:

The authors have presented a phylogeny and used it in their analysis but there is no description or citation for how that phylogeny was derived.

Line 198:

The discussion around oxyR and ahpC and ahpD would benefit from greater detail. Are the authors suggesting that oxyR is functional in L3? Or are they just commenting that it is simply upregulated transcriptionally but still non-functional? Similarly, many variants are known to cause ahpC over-expression. Are there any variants upstream of ahpC which may account for this finding in particular?

Lines 234-237:

I like the comparison with other boxes, but it is unclear what the sample size is for each of these other sigma factor binding sites when compared with sigA. Is the lack of effect because the fraction of boxes affected is smaller or because there is a smaller sample size?

Line 256: The data presented in figure 3b is presented as

Figure 3b:

The data presented here clearly show that there are more C and G sites mutated than A and T overall, however this analysis is not corrected for the GC content of *M. tuberculosis* in which G/C sites are over-abundant. If the authors suspect that this skewing is beyond what would be expected based on GC content they should perform a statistical test incorporating this bias into the expected ratio of sites.

Figure 4a:

The figure is largely uninterpretable because the symbols are overlapping and it isn't possible to see the actual distribution of triangles and circles. I suggest breaking out each lineage into one set of data for expression of genes with new boxes and one set for expression of genes with disrupted boxes. The data as presented do not clearly support their conclusion. It is also not clear what they mean by 'nearby genes' in this analysis. Would this be genes on either side of the box? And are these boxes confined to intergenic regions or within genes?

Figure 4B:

The figure legend text would do well to be stated in the main text. The authors bring up the over-expression of ahpCD in L3 but then it is not mentioned again directly in the text.

Line 282:

I think this is worded a little strongly. As many as 26% could be accounted for by new boxes, it is not definitive that these are causal.

Table 1:

It would help a reader not deeply familiar with the methylation literature to know the motifs you used and how you assigned them to each methyltransferase in the table. As a result of this, it is also not clear how many sites are in the denominator for each of these frequencies. Also the term hemimethylated specifically means methylated on one strand. Is this the authors intended statement or do they mean partially methylated?

Figure 5b:

Perhaps the authors could perform a statistical test asking whether the number of genes in the top left panel showing over-expression is statistically unlikely?

Figure 5c:

The figure refers to a Rv3727 but in the text Rv3272 is mentioned. Is this a typo? Also Rv0898c and Rv3616c are listed twice in the box for sigA alternative motifs.

Figure 6:

The lack of a complemented strain makes it difficult to interpret if this transcriptional effect is due to hsdM mutation. Given that none of the genes have hsdM dependent methylation near them, I agree that the effects are indirect and thus there is not much evidence that any change other than hsdS.1 is due to their deletion and this itself may be a polar effect. It is not clear what conclusion to draw from this figure or how it adds to our understanding of hsdM function.

Methods:

Line 541-547: The authors should include a specific list of the genes they excluded in the analysis due to deletion or repetitive genes. And they should clarify how they identified and defined deleted genes and how they specifically accounted for this. Was the gene removed from all analyses if deleted in a single isolate? Or just in analyses including that isolate?

Lines 562-592:

It would help to clarify that these are three separate tests being performed. Although in supplementary figure 3 there is a red line connection panel a and b which I do not understand. Is the poison being used somehow in the first permutation test as well?

Line 620-623:

The calculation of dN/dS can be performed in many ways. This description is insufficient for others to replicate the authors calculations. If it is in the same as reference 42, they should cite this in their methods section.

Reviewers' comments:

Reviewer #1 (Remarks to the Author):

The authors performed transcriptome, genome and methylome analysis to systematically screen the differentially expressed genes between MTBC clades, and further discovered that genomic variants and their disruption of methyltransferase have critical role on expression regulation. They also searched a lot of public datasets to gain a full perspective on their main conclusions. The findings obtained by the authors are informative to describe the relationship among genome, transcriptome and methylome from MTBC strains. However, as some bioinformatics papers, the manuscript is not well organized since the authors tried addressing too many results and message. The following concerns should be addressed.

Q1. The gene expression and methylation states are real-time, unstable, and they often varied with bacterial growth. The authors should performed the RNA and DNA extractions using the same batch of cells at the same growth timepoint/state. Otherwise, it is not reasonable to conduct comparative analyses among transcriptome, methylome and genome.

As far as the authors know, there is no evidence in TB that methylation states are unstable over short timescales, and no reason to expect them to be, given that the methylase genes show no evidence for differential expression over short timescales or of anything that might make them phase-variable. The phylogenetic data and past studies [PMCID: PMC5760664 and PMCID: PMC4737169] suggests they are stably maintained for long periods with very occasional mutations or deletions. Therefore there is no requirement to extract the RNA and DNA from exactly the same state.

Q2. Line 123-128: the authors remove strain N1177 from the analysis based on the PCA result and an *rpoB* mutation. The author said that *rpoB* mutation (D435Y) affected the RNA-polymerase and transcription. Pls provide the related reference. In addition, *rpoB* mutation is a type of common one in TB strains, so the authors should search for the *rpoB* mutation in all the MTBC strains and remove them from the study.

Four *rpoB* nonsynonymous mutations were found in the dataset, all of them in L6 strains (T350I, S388L, D435Y and E639D). D435Y is the only one conferring RIF-resistance according to published catalogs and the only falling in the active center and in the rifampicin resistance determining region (<https://doi.org/10.1016/j.cmi.2016.09.006>). S388L and E639D were common to all L6 strains, so they could not be involved in intralinear variability of gene expression. The T350I mutation was found only in N0091. While there is no information on the effect of this mutation, the position is not in the RNA polymerase active center so it is unlikely to be involved to changes in transcription rates.

We have added new references supporting this fact and modified the main text in lines 127-128: "After the initial analysis, we realized that N1177 harbours a mutation in the *rpoB* gene (D435Y) that confers resistance to rifampicin. As mutations affecting the RNA-polymerase could

have pleiotropic effects^{23–26} and hence alter transcription at a genome-wide level, it is not surprising that N1177 does not cluster together with the other L6 strains.”

Q3. Line 119-128: the explanation for preservation of stain N0031 in the study is inadequate. If the expression of stain N0031 is different from other L2 strains, the authors should also remove the strain like strain N1177.

We respectfully disagree with the reviewer. N0031 is part of the natural diversity of the L2, it falls in an early branch (also known as proto-Beijing) and as such its transcriptional profile has to be taken into account when analyzing core transcriptional differences in L2.

Q4. Line 280-285: the authors described that 26% transcriptional variability found across the MTBC main clades was due to single point mutations. As one of the main conclusions in this study, I recommend that the authors validate some altered gene expressions by qPCR.

We thank the reviewer for the suggestion. Instead of qPCR, which will take time and resources given the slow growth of MTB, we have used a second RNAseq dataset generated in a different laboratory and previously published (labelled as Rose *et al.* data in the figure) and with slightly different culture conditions for L1, L2 and H37Rv strains (no pyruvate added in the media, see PMID: 24115728 for details).

In the new dataset we have analyzed the consistency of the findings reported in the submitted version (labelled as This study data) for those genes with a new box and altered transcription levels. The results are 100% congruent. Each time a new pribnow box appears we see a higher gene expression in the gene and lineage affected:

Figure 1 Effect of the new Pribnow boxes over gene expression tested in independent datasets. New Pribnow boxes generated by point mutations increase the expression of nearby genes in the strains in which the new box appears. The observed effect is independent of the RNAseq dataset used. When the new Pribnow box appears in the noncoding strand, the

increase in expression is observed in the antisense RNA (denoted in the x-axis as ‘_as’). From the Rose et al. dataset⁹ strains N0145 (L2), N0153 (L1) and H37Rv (L4) were used.

We have now included these results and the figure as part of supplementary material and mentioned it in the following line of the text:

Line 29-303: “To corroborate our results, data for L1 and L2 strains along with H37Rv grown in a different laboratory were obtained from previously published work¹⁹. The same pipeline explained in the Methods section was applied to this new dataset. We see the same expression trends in those genes in which we originally linked a higher transcription rate to a lineage mutation generating a new pribnow box (Fig S5).”

Q5. L365-384: the authors described that two methyltransferases, MamA and MamB could influence the expression of some genes through the overlap between the methyltransferase recognition motifs and SigA recognition motifs. However, this is not the case for HsdM. It is better if the authors could further describe the pattern of various HsdM recognition motif sequences (GATN₄RTAC), such as GATN₄ATAC and GATN₄GTAC.

Following the reviewer’s advice, we have reviewed the impact of HsdM on gene expression, splitting the analysis depending on the two alternative motifs methylated. We have selected lineage 4 strains, as we have two strains with HsdM active and one with the methyltransferase inactive. Following the same approximation used in the main text, we have tested the log₂ fold-change in expression for genes having their SigA motifs differentially methylated. What we observe (figure 2) is that, even when splitting the motif GATN₄RTAC in its two different alternatives, there is not a clear effect of the DM over genes expression. In opposition to MamA and MamB, the DM mediated by HsdM seems to not affect the gene expression in a consistent manner.

Figure 2. Gene expression differences between SigA recognition motifs differentially methylated by HsdM. Red line marks a 0 fold-change in gene expression (no differences). The expression of each gene was tested in both situations, methylated and non-methylated strains, in lineage 4.

The results are expected given that the 'R' base in the GATN₄RTAC motif do not overlap with the SigA motif in most of the cases. And in the cases that it does, it overlaps with the 'N' bases of the TANNNT (SigA) motif. Thus, we think that it is unlikely that the effect of DM over GATN₄ATAC and GATN₄GTAC could be different.

Figure 3. Different overlapping patterns found between SigA recognition motifs and the HsdM methylated motifs. The red adenines are the differentially methylated ones, while the bold purple purines are the bases which can be either 'A' or 'G'.

Reviewer #2 (Remarks to the Author):

The manuscript by Chiner-Oms and colleagues describes a transcriptomics analysis of multiple strains of Mycobacterium tuberculosis from distinct evolutionary lineages. The authors present a substantial amount of data that includes a comprehensive transcriptomic analysis that encompasses 19 M. tuberculosis strains from six lineages, global methylation profiling of these strains to explore links between gene expression and differential methylation of promoter elements, and analysis of a methyltransferase mutant vs wild type. The authors present evidence that lineages of M. tuberculosis have shared transcriptomic signatures, and hypothesise that this transcriptional diversity links to phenotypic diversity and ultimate success of the lineages.

I have some comments I would like to see addressed.

Q6. A caveat of the study is that only a single in vitro growth condition was studied, namely exponential growth in laboratory media, and this is acknowledged (line 391). Looking through the detail of the manuscript I wasn't clear exactly at what point in the growth curve the strains were harvested.

We want to thank the reviewer for pointing out this fact. The different OD₆₀₀ stated in the submitted version was a typo.

All cultures were grown in the same medium to the same growth state. The cultures were started at OD=0.01 and harvested at an OD around 0.5. The specific growth rate (shown as doubling time) of each individual strain is shown below (figure 4). Small differences in growth rates are expected when one deals with such a heterogeneous group of Mtb strains.

We have rewritten the methods and results sections to clearly state that DNA and RNA was collected at an OD₆₀₀ of 0.5-0.7 (see lines 541 and 549).

Figure 4: Growth rates for all the samples included in this study

Q7. It would be necessary to describe whether the growth rates of the strains were similar under the in vitro conditions used; otherwise it could be argued that the transcriptional differences across strains are simply a function of in vitro growth rate differences. For example, it is mentioned that pyruvate was added to media “to account for strains with potential pyruvate kinase mutations”; does this mean pyruvate was the sole carbon source used? Or was glucose or glycerol also present? Choice of carbon source by strains may affect growth kinetics and hence global transcriptional activity.

All the strains were grown in 7H9 OADC with 30 mM pyruvate as mentioned in the manuscript and the growth rates were similar (see Figure 4 above). The OADC supplement contains glucose, so the two main carbon sources are glucose and pyruvate. It is possible that different strains might have different preferences for carbon sources yet this is impossible to account for. By using a standard medium with the addition of pyruvate for all, we are maintaining identical growth conditions.

Q8. Line 119-128: The authors show that the N0031 strain does not match the transcriptional profile of other strains in the lineage because of a mutation in a master transcriptional regulator, *dosR*. Likewise, the N177 strain did not cluster with other L6 strains due to a mutation in *rpoS*, conferring rifampicin resistance. Hence the results from both of these isolates show that strains within a lineage can have a divergent transcriptional profile from other strains, and that this may particularly be the case when strains are drug-resistant. Perhaps this latter point, with relevance to drug resistance, could be more explicitly made in text.

We agree with the reviewer’s suggestion. There are lineage trends that can be substantially altered by single point mutations as the two mentioned. We have now expanded the discussion on this topic. New main text:

Lines 467-469: “Now, we show that even single point mutations may totally change the transcriptional profile of a strain. An example is strain N1177, which carries a single mutation in the *rpoB* gene conferring rifampicin resistance which modified the transcriptional levels of multiple genes. Likewise, a single mutation generating a new SigA recognition motif increases the expression of the DosR regulon in the three Lineage 2 Beijing strains but not in the basal Lineage 2 strains (Fig 1a, ¹⁹).”

As further evidence to support the authors’ hypothesis, I wondered whether a global analysis of the lineages would reveal significantly more mutations in genes encoding transcriptional regulators? For example, it has been shown that mutations in the PhoPR system of *M. tuberculosis* can affect the expression of genes which may be involved in the distinct host preference of *M. tuberculosis* complex members. Similarly, serine-threonine protein kinases show differences across *M. tuberculosis* strains that impact on global gene regulation. Might such master regulators harbour greater genetic diversity than would be expected by chance, and hence play into transcriptional plasticity across lineages? I realise that the authors were

focusing on 19 strains for transcriptome analyses, but augmenting this with a global analysis of available *M. tuberculosis* genome sequences would fit well within the manuscript and may provide further evidence that would support their hypothesis.

The reviewer is right and we share some of his thoughts regarding the conservation degree of the regulatory network. We have studied the conservation of transcriptional factors in a previous publication (PMCID: PMC5830583). In that publication we show, using hundreds of strains, that transcription factors are not conserved across lineages (including deletions affecting complete lineages). In an even more recent publication (in press) we have shown by inspecting thousands of strains that mutations in *phoR* are selected in the evolution of the MTBC and thus likely impact transcriptional profiles by controlling the master regulator *phoP*. Unfortunately it is not feasible to do transcriptomics on thousands of strains. This is now stated both in the introduction and the discussion.

Line 59: "For some cases, the genetic bases of the expression differences are known. We have previously shown that MTBC regulatory networks vary across strains and lineages, with several transcription factors carrying mutations that potentially impair regulatory function¹⁷."

Lines 467-169: "Each main lineage is defined by a transcriptomic landscape, that clearly separates it from the rest of the lineages. We have shown before that transcriptional regulators are not conserved across lineages¹⁷. Now, we show that even single point mutations may totally change the transcriptional profile of a strain."

Q9. Line 195-201, and Fig4c: Lineage 3 strains were shown to have increased expression of *ahpC-ahpD-oxyR*, and that this may be linked to the creation of a new Pribnow box. It would have been good to have seen this experimentally verified by plasmid reporter constructs.

Thank you for this suggestion. RNA seq results were done in duplicates and are quantitative. In addition, the fact that we see increased transcript levels for the full operon (*ahpC-ahpD*) increases the confidence in this being a legitimate increase in expression.

Q10. Line 229: "...we randomly introduced all the mutations observed in the genome...". By 'the genome', do the authors mean mutations across all lineages, including *M. africanum*?

Yes, we meant all the mutations observed in the 19 strains. We have clarified this point in the text lines 233-234: "...we randomly introduced all the genomic mutations observed in the 19 strains and repeated the process 1,000 times...".

Q11. Line 398-400: "...the natural variability of HsdM found in the MTBC suggests that there may be some biological relevance associated with this protein." What is meant by 'natural

variability', and 'biological relevance' here? I think more precision would be needed here as to why *hdsM* was selected for mutant construction and transcriptome analyses.

The reviewer is right about the lack of a proper explanation on why we chose HsdM. We have now replaced the terms “natural variability” and “biological relevance” with an expanded explanation. We based our decision on the number of stop codons observed for *hdsM* despite having a low dN/dS (low accumulation of aminoacid changes). The later suggests that the transferase is functional and under purifying selection, the former suggests that the methylation activity has been lost independently in different branches of the phylogeny.

Lines 372-377: “Despite gene-wide conservation of the methyltransferases we observed the accumulation of functional mutations in the form of new stop codons. For example *hdsM* accumulates 5 stop codons in different parts of the phylogeny suggesting that either the gene is under weak selection (contradicting by the low dN/dS observed) or that specific mutations on the gene have been selected during evolution even though we don't observe any impact on expression profiles of unmethylated strains.”

Lines 425-433: “While minor effects on gene expression were found linked to MamA and MamB (Fig 5b) we were surprised that no effect could be linked to HsdM, particularly as HsdM is the one that has accumulated more stop codons during the evolution of the MTBC (Table S7). This suggests that it is either under weak selection, randomly accumulating inactivating mutations, or that it has a functional role and specific mutations in HsdM have been selected in different parts of the MTBC phylogeny. To discriminate between these two possibilities we mimicked a stop codon mutant for HsdM by deleting the *hdsM* gene in a N1283 background (L4). Compared to the other two L4 strains in the transcriptome dataset HsdM is fully functional in N1283 (Table 1) which allow us to compare it to transcriptomic profiles of unmethylated MTBC strains.”

Q12. Line 440-1: “This is another proof that lineages of MTBC likely reflect adaptation to different human populations”. It would be better to say “This provides further evidence that lineages of MTBC likely reflect adaptation to different human populations”.

Done, lines 476-477

Reviewer #3 (Remarks to the Author):

Clinical isolates of *Mycobacterium tuberculosis* (Mtb) have been repeatedly shown to have meaningful phenotypic heterogeneity in virulence, however our current understanding of the mechanisms driving these differences is incomplete. Here Chinar-Oms et al. utilize transcriptomics and whole-genome methylation analysis of a diverse set of clinical isolates to track the divergence of Mtb expression profiles across its evolutionary history. In the process,

they present a framework for analyzing transcriptome changes across the phylogenetic tree which allows them to look for mutations and methylation pattern alterations which correspond with altered gene expression.

Overall the manuscript provides a wealth of new data which was not previously available, however there are several areas where the authors would benefit from further clarifying their methods and supporting their conclusions with statistical tests.

Q13. In particular, the authors focus on ascribing genetic changes in pribnow boxes to alterations in gene expression, however their analysis linking these two phenomena is not supported by any statistical test. Clearly not all mutations in pribnow box sequences identified by sequence alone will mediate a change in gene expression, however if they want to ascribe these mutations to the altered expression then there must be some additional information provided.

We agree with the reviewer and we have now performed several statistical tests to support our hypothesis. First, we have compared the distribution of log₂ fold-changes in expression for genes having boxes in their upstream regions (those from figure 4a). A wilcoxon test shows that new boxes increase the expression of downstream genes while disrupted boxes decrease the expression (wilcoxon test, p-value = 5.37-E09).

In addition, we have found a significant enrichment of genes being identified as upregulated in the PDEG analysis and the appearance of a new Pribnow box in the upstream regions of these genes (chi-squared test, p-value = 2.78E-09).

In addition, we have used an external dataset to corroborate our results (see reviewer#2, Q4)

With these new results, we have modified the main text:

Figure caption 4a: "Effect of the new/disrupted Pribnow boxes over the expression of downstream genes. New boxes tend to upregulate gene expression while disrupted boxes tend to downregulate transcription (wilcoxon test, p-value = 5.37-E09)."

Lines 282-291:"We took into account only those mutations affecting the clades defined previously in the PDEG as the analysis of individual strains could lead to inconsistent results due to the lack of statistical power. First, new pribnow boxes are overrepresented among upregulated PDEG genes (chi-squared test, p-value = 2.78E-09). Second, when taking into account all genes, not just PDEG, we always observed higher expression of genes with a new pribnow box due to a mutation compared to the closest relatives without the mutation. Conversely genes losing the pribnow box because of a mutation have lower expression (Fig 4a, wilcoxon test, p-value = 5.37-E09). A clear example is the observed overexpression of *oxyR* in L3 strains, potentially linked to a mutation (G2726105A) that creates a new Pribnow box (Fig 3c)."

Q14. The global analysis of methylation among these clinical isolates, and corresponding identification of mutations in methyltransferase machinery is also very interesting however the data presented on the *hsdM* mutant do not appreciably add to the paper.

We respectfully disagree with the reviewer. The *hsdM* mutant adds evidence about secondary mechanisms of gene expression control by methyltransferases. This data is important to be reported to raise awareness among many research groups that will try to identify direct links between methylation and altered transcription. We have also now made it more clear why we selected *hsdM* (please see response to Reviewer #2, Q11).

Q15. Finally, the authors focus almost exclusively on differences among the strains at the level of somewhat arbitrarily designated lineages. To that point, the most divergent lineage 2 isolate in fact doesn't appear much like other lineage 2 isolates transcriptionally. Given the rich nature of this dataset, it would be illuminating to report what fraction of transcriptional variation can be accounted for by the lineage designation as opposed to differences among individual strains.

We disagree that the designation of the lineages is arbitrary. The MTBC lineages have been well established using different genetic markers (deletions, SNPs, genomes). In addition, population genetics approaches to delineate genetically related groups have shown that strains within lineages consistently cluster much closer than with strains from other lineages showing that the separation is scientifically sound (see Coll 2014, PMID: 25176035 and Comas2013-Figure1b, PMID: PMC3800747 , for an example).

We however agree with the reviewer that it could be interesting to get insights into the intra- vs. inter-lineage gene expression differences. Thus, using the complete transcriptomic data for each sample, we have calculated the pairwise Euclidean distance between samples as a measure of the transcriptomic variability between them. Although conclusions from such a limited number of samples will not reflect the complete MTBC scenario, the intra-lineage transcriptomic variability resembles a lot the within-lineage pairwise SNP distance pattern (see Coscolla&Gagneux 2014, PMID: PMC4314449). We observed that the intra-lineage diversity is lower than the inter-lineage diversity, supporting our decision to group the strains according to their phylogenetic origin.

We have added a statement in the Result section and a Supplementary figure (Figure 5) pointing to these results.

Lines 117-120: "Furthermore, the intra-lineage genome-wide expression distance between samples is lower than the inter-lineage distances (Fig S2), supporting the idea that samples from the same lineage have a profile more similar to each other than with samples from other lineages. "

Figure 5. Intra- and inter-lineage pairwise euclidean distance distribution, calculated from the complete transcriptomic data. This pattern of variability resembles a lot the within-lineage pairwise SNP distance pattern (see PMID: PMC4314449).

Q16. Abstract:

Line 32: "The alterations in gene expression" sentence is confusingly worded.

Changed to 'The changes in gene expression'. Line 32 :

Q17. Introduction:

Line 42:

Monomorphic seems like a strange term to use. Perhaps clonal or non-recombining.

Done. 'monomorphic' changed to 'non-recombining' in line 42.

Q18. Results:

Line 107:

It is not clear to me why you would merge the data from two replicates rather than treat them as actual replicates in further analyses.

The correlation between the transcriptomic profile of the replicates is almost 1 (range 0.9996 - 0.9999). In figure 6, we show the PCA of the transcriptomic profiles, without merging the replicates. Replicates of the same strain overlap in the PCA plot, showing the high similarity between the transcriptomic profile of each run. As we have focused in the inter-lineage variation, the minimum variability found between replicates of the same strain will not affect the gene expression results. So, we decided to merge the replicates to increase the number of read

counts per gene, necessary for fine-grained subsequent analyses. Anyway, the differential expression analysis was performed by using both approaches (with and without merging replicates) and the results were the same.

Figure 6. PCA plot obtained from the complete transcriptomic profile of each sample, without merging the replicates.

Q19. Line 110:

It isn't clear if the PCA was performed on data which is coverage normalized or whether it was just log2 transformed. It should be.

The reviewer is right. The PCA was performed on data previously normalized and log2 scaled by using the rlog function in the DESeq2 package. This is now stated in the Methods section, in lines 564-565 :”The PCA and the hierarchical clustering were performed by previously normalizing the count data across samples and scaling it into a log2 scale, by using the rlog function from the DESeq2 package.”

Q20. Line 127:

The authors state sample N1177 is removed because rpoB mutant which is probably messing up all sorts of stuff. This isolate is still included in Figure 2a and it is unclear if it was incorporated into the calculations for Figure 2b.

Sample N1177 was included in figure 2a to show the complete phylogenetic picture, however it was not used for further analyses (neither the calculations for figure 2b). We have modified the figure 2a and the figure caption so the readers could be aware of this.

Fig2a caption: "Number of genes differentially expressed (red up, blue down) in each of the main branches of the MTBC phylogeny. The phylogeny was constructed using Illumina sequencing data, the Maximum-Likelihood algorithm and a bootstrapping of 1,000 replicates. Sample N1177 is included to shown the complete phylogenetic picture, but it was not included for further analyses."

Q21. Line 143:

Change contrarily into "In contrast"

Ok, changed in line 145

Q22. Line 155:

The authors have presented a phylogeny and used it in their analysis but there is no description or citation for how that phylogeny was derived.

Ok, this information is now included in the methods section.

Lines 601-603 : "An MTBC phylogeny was calculated by using the RAxML program⁶⁴ with the GTRCATI model of evolution and represented with the iTOL software⁶⁵."

Q23. Line 198:

The discussion around *oxyR* and *ahpC* and *ahpD* would benefit from greater detail. Are the authors suggesting that *oxyR* is functional in L3? Or are they just commenting that it is simply upregulated transcriptionally but still non-functional? Similarly, many variants are known to cause *ahpC* over-expression. Are there any variants upstream of *ahpC* which may account for this finding in particular?

We agree with the reviewer that we could add more information to the *oxyR-ahpC* and *ahpD* overexpression. We do not think that *OxyR* is functional in L3, as it has been reported as being inactive in the MTBC due to several deletions in comparison with other mycobacteria (PMID: 8596438). We think that only its transcription rate is affected but not its functionality. We have modified line 203 to reflect this : "It has been previously reported that *oxyR* is inactivated in H37Rv, BCG, *M. africanum* and *M. microtti* due to several deletions that affect its translation³¹."

Regarding the *ahpC* variants, we found 2 more variants in the upstream *ahpC* region. We have now added this results to the supp text : "We have found two other variants in the *oxyR-ahpC* intergenic regions, G2726051A and C2726121T. None of them are present in L3 strains and are not involved in the creation of new TANNNT motifs. G2726051A is present in all the L1 strains, and we did not see an increase in *ahpC* expression levels in this clade. C2726121T is only found in the N1177 strain (L6), which is the one that we have omitted from our analyses as it harbours the *rpoB* mutation."

Q24. Lines 234-237:

I like the comparison with other boxes, but it is unclear what the sample size is for each of these other sigma factor binding sites when compared with sigA. Is the lack of effect because the fraction of boxes affected is smaller or because there is a smaller sample size?

We think that the reviewer has raised an interesting question. We have no doubt that sample size is not affecting the permutation test results. SigA and SigE recognition motifs are found in the genome in a similar amount (table 1), however the number of new/disrupted motifs are highly different, as well as the results derived from the permutation test.

Sigma factor	Motifs lost			Motifs gained			Total in ancestor genome
	N	Z-score	Probability	N	Z-score	Probability	
SigA	81 (0.5%)	2.24	0.015	683 (4%)	15.59	0	15,283
SigE	94 (0.5%)	-6.93	2.10E-12	119 (0.7%)	-7.80	3.09E-15	15,797
SigG	91 (0.8%)	-5.67	7.16E-09	60 (0.8%)	-4.39	5.66E-06	10,818
SigJ	19 (1%)	-0.09	0.46	11 (1%)	-2.55	0.005	1,009

Table 1. Number of new and disrupted motifs due to the genomic mutations found in the 19 strains dataset. Last column shows the number of motifs found in the MTBC ancestor genome.

In addition, when we randomly reintroduced all the genomic mutations observed in the 19 strains, we observed the gain or loss of the sigma recognition motifs due to stochastic processes, independent of the sample size. The permutation test (1,000 iterations) show a different profile for SigA and the other sigma factors recognition motifs (SigE, SigG and SigJ). While reading the reviewers question, we realized that it could be interesting to add more extra info about this analysis in the publication. We have now stated explicitly that, while the new SigA motifs are higher than those expected by stochastic processes, the number of new/disrupted motifs in SigG and SigE sigma factors are lower than those expected by chance. We have now included a new Supplementary figure (figure 7) with these results and modified the main text in lines 239-242: "In addition, when we repeated this permutation test for other sigma factors' -10 consensus sequences such as SigE (cGTT), SigG (CGANCA) and SigJ (CGTCCT)⁴⁰, we observe the opposite pattern (Fig S3, Supplementary text). Our observations support the hypothesis that new SigA boxes are maintained by selection and not genetic drift." We have also included a new Supplementary text (section "Natural selection shapes sigma factor's recognition motifs abundance") discussing these results.

Figure 7. Heterogeneous impact of the observed mutations over the sigma factors recognition motifs. New SigA recognition motifs are introduced in the MTBC genome at a higher rate than expected by chance, pointing to the effect of non-stochastic processes in the accumulation of new TANNNT motifs. In contrast, SigG and SigE recognition motifs are created/disrupted at a lower rate than expected by chance.

Q25. Line 256: The data presented in figure 3b is presented as

Please can you clarify the comment?

Q26. Figure 3b:

The data presented here clearly show that there are more C and G sites mutated than A and T overall, however this analysis is not corrected for the GC content of *M. tuberculosis* in which G/C sites are over-abundant. If the authors suspect that this skewing is beyond what would be expected based on GC content they should perform a statistical test incorporating this bias into the expected ratio of sites.

The reviewer is correct. In Hershberg *et al.* 2008 (PMCID: PMC2936535) the authors demonstrate that mutations are biased towards AT in the MTBC, taking into account the GC

content and correcting for it. We have now corrected the GC bias by following the approach published in this publication:

From Hershberg *et al.* 2008 "In order to account for the unequal nucleotide content [...] we normalized the counts of the mutations from A/T to G/C, C/G, or T/A by multiplying them by $\frac{\#GCsites}{\#ATsites}$, where $\#GCsites$ and $\#ATsites$ are the current genome wide number of GC or AT sites".

Following this approach instead of the absolute number of mutations we have now plotted the GC corrected relative abundance. The figure caption has been modified to account for this fact:

" **b**, Mutation bias towards new A and T alleles inferred from 235,212 substitution obtained from 4,595 clinical samples of the MTBC and normalized by GC content as in ²⁰."

We have also modified the main text to make more the Hershberg *et al.* results. Line 259:" It is known that there is a bias towards TA substitutions in bacteria, even in the case of rich GC-rich genomes such as the MTBC case²⁰."

Q27. Figure 4a:

The figure is largely uninterpretable because the symbols are overlapping and it isn't possible to see the actual distribution of triangles and circles. I suggest breaking out each lineage into one set of data for expression of genes with new boxes and one set for expression of genes with disrupted boxes. The data as presented do not clearly support their conclusion. It is also not clear what they mean by 'nearby genes' in this analysis. Would this be genes on either side of the box? And are these boxes confined to intergenic regions or within genes?

Figure 4a has been now split by the effect of the mutation over the box, as suggested by the reviewer. In addition, we have added boxplots to show that almost all the changes in gene expression vary according to the effect of the mutation over the Pribnow boxes (new boxes increase expression, while disrupted boxes decrease it). Note that changes reported as being detected in the PDEG analysis have undergone the initial filters of $\text{adj-pvalue} < 0.05$ and $\log_2 \text{fold-change} > 1.5$. Thus, subtle increases/decreases in expression were not classified as PDEG, although they could be driven by Pribnow box mutations. We have modified the figure 4a caption text for clarification. "Blue circles represent those changes in expression detected in the PDEG analysis ($\text{adj-pval} < 0.05$, $\log_2 \text{fold-change}$ in expression > 1.5). Red circles represent subtle changes in gene expression, thus not identified by the PDEG analyses."

By nearby genes we meant genes downstream of the mutation. Mutations can appear either in the intergenic region of these genes or in the coding regions. We have now changed the word

'nearby' for downstream/upstream across the main text, as we agree with the reviewer that the term was confusing.

Q28. Figure 4B:

The figure legend text would do well to be stated in the main text. The authors bring up the over-expression of *ahpCD* in L3 but then it is not mentioned again directly in the text.

Done. We have moved this to the main text. Lines 289-291: "A clear example is the observed overexpression of *oxyR* in L3 strains, potentially linked to a mutation (G2726105A) that creates a new Pribnow box (Fig 3c)."

Q29. Line 282:

I think this is worded a little strongly. As many as 26% could be accounted for by new boxes, it is not definitive that these are causal.

Done. We have toned-down this sentence. Line 295: "meaning that ~26% of the transcriptional variability found across the MTBC main clades could be linked to single point mutations."

Q30. Table 1:

It would help a reader not deeply familiar with the methylation literature to know the motifs you used and how you assigned them to each methyltransferase in the table. As a result of this, it is also not clear how many sites are in the denominator for each of these frequencies. Also the term hemimethylated specifically means methylated on one strand. Is this the authors intended statement or do they mean partially methylated?

We have specified the motifs detected in the main text (line 316). We have also added the number of occurrences of these motifs in the table, and modified the table caption:

Line 315-316: "Consistent with previous reports, we identified three main methylated motifs (CTCCAG, GATNNRTAC and CACGCAG) in almost all the samples."

Table caption: "Table 1. Methylation profile of the 19 MTBC samples. Summary of the methylation report derived from the PacBio sequencing for each of the three main methyltransferases. For each motif, the methyltransferase involved in its methylation and the number of occurrences identified in the genome are reported in the header. In almost all samples, all the motifs identified are methylated. In those samples having less than 80% of the motifs methylated, nonsynonymous variants were identified in the relevant methyltransferase coding genes."

We have also included the literature describing the methyltransferases that recognize each of these motifs.

Lines 329-331: "These motifs have been previously reported to be recognized by three main MTBC methyltransferases MamA, HsdM/HsdS.1 and MamB⁴³⁻⁴⁵."

In Methods there is a section explaining how the PacBio data is used to infer the number of methylated motifs in each sample.

The reviewer is correct; we meant partially methylated, not hemimethylated, and we have corrected this.

Q31. Figure 5b:

Perhaps the authors could perform a statistical test asking whether the number of genes in the top left panel showing over-expression is statistically unlikely?

We have now compared for Lineage 1 and for Lineage 2 the mean in log₂ fold-change expression between genes with and without methylation overlapping the sigA_motif. In both cases the result is significant (Wilcoxon test p-val = 0.04 for Lineage 1 and 0.003 for Lineage 2). However, the number of genes tested in both categories (without overlap versus with overlap) are very different, so we prefer not to include these results in the main text.

Q32. Figure 5c:

The figure refers to a Rv3727 but in the text Rv3272 is mentioned. Is this a typo? Also Rv0898c and Rv3616c are listed twice in the box for sigA alternative motifs.

Yes, these are typos. They have been now corrected.

Q33. Figure 6:

The lack of a complemented strain makes it difficult to interpret if this transcriptional effect is due to hsdM mutation. Given that none of the genes have hsdM dependent methylation near them, I agree that the effects are indirect and thus there is not much evidence that any change other than hsdS.1 is due to their deletion and this itself may be a polar effect. It is not clear what conclusion to draw from this figure or how it adds to our understanding of hsdM function.

We are also unsure about the interpretation of these results. However we think it is important to report that most of the methylation effects in this particular case are indirect. We think the data will help research groups looking for mechanisms and targets of methylation.

Q34. Methods:

Line 541-547: The authors should include a specific list of the genes they excluded in the analysis due to deletion or repetitive genes. And they should clarify how they identified and defined deleted genes and how they specifically accounted for this. Was the gene removed from all analyses if deleted in a single isolate? Or just in analyses including that isolate?

We have now included this list of genes. We identified those genes of the reference genome (H37Rv) that had gaps in the coverage when mapping the PacBio sequencing data on them. The deleted genes were removed only in those analyses in which the strain lacking these genes were involved, not in all the analyses. We have specified the approach in methods and included a supp data with the genes removed from the analyses.

Lines 580-583: "Genes with deletions in each of the groups were not taken into account in the pairwise comparisons, as they result in false positive signals. These genes were identified by

mapping long-reads obtained from PacBio sequencing against the H37Rv reference genome, and assessing the genomic coverage (Table S9).”

Q35. Lines 562-592:

It would help to clarify that these are three separate tests being performed. Although in supplementary figure 3 there is a red line connection panel a and b which I do not understand. Is the poisson being used somehow in the first permutation test as well?

The reviewer is right that there are three independent tests. The Poisson distribution was used only in the first test. The red line meant that both methods were used to assess the probability of the observed motifs due to stochastic processes. We have remade the figure for clarification purposes and include a new line in the Methods section.

Line 609-610:”Three independent tests were performed to calculate the probability of the observed new/disrupted SigA recognition motifs by chance.”

Q36. Line 620-623:

The calculation of dN/dS can be performed in many ways. This description is insufficient for others to replicate the authors calculations. If it is in the same as reference 42, they should cite this in their methods section.

The reviewer is right and we have now expanded the explanation of the dN/dS calculation methods. Line 664-669:”dN/dS values for the methyltransferases were calculated as described previously in ³⁷ . Briefly, by using the observed nonsynonymous and synonymous variants in these genes, and the potential synonymous and nonsynonymous substitution sites for each gene (calculated using the SNAP tool⁶⁹ and without distance correction), the dN/dS for each gene was:

$$\frac{\text{Nonsynonymous variants} / \text{Nonsynonymous sites}}{\text{Synonymous variants} / \text{Synonymous sites}}$$

REVIEWERS' COMMENTS:

Reviewer #1 (Remarks to the Author):

The responses for Q1, 2, 4 and 5 are acceptable. As for Q3, the authors need to provide some literature to support their conclusion: the transcriptome of proto-Beijing strains is different from that of other Beijing strains. Also, the author should explain why strain N0031 is close to L4 strains based on the PCA plot?

Reviewer #2 (Remarks to the Author):

The authors have addressed all my comments and clarified the issues I raised. As such I have no further comments and am happy with the revised manuscript.

Reviewer #3 (Remarks to the Author):

The authors have addressed my concerns.

Reviewer #1 (Remarks to the Author):

The responses for Q1, 2, 4 and 5 are acceptable. As for Q3, the authors need to provide some literature to support their conclusion: the transcriptome of proto-Beijing strains is different from that of other Beijing strains. Also, the author should explain why strain N0031 is close to L4 strains based on the PCA plot?

We want to thank the reviewer for highlighting this point. We noticed that we have not offered a detailed answer to this question in the previous interaction.

It has been previously reported (mainly by Rose *et al.*, 2013 (PMID: 24115728) but also by others) that the proto-Beijing clade has a different transcriptomic profile in comparison with the Beijing L2 strains. These differences are due to an overexpression of the *dosR* regulon in the Beijing strains (Reed *et al.*, 2007 (PMID: 17237171), Homolka *et al.*, 2010 (PMID: 20628579)). In fact, Rose *et al.*, showed that the expression pattern of the *dosR* regulon in N0031 (the same strain that we used in our work) resembles more the expression of the regulon in other lineages than in Beijing L2 strains (Rose *et al.*, 2013 (PMID: 24115728), see figure 2A). We have modified the text to clarify this point and we have made more obvious the reference to the previous literature.

Regarding the concern of the reviewer about our PCA plot, the reviewer should take into account that we are just plotting two principal components, which account for 44% of the variance. Actually, if we plot more components we see that N0031 separates both from the rest of L2 and from L4 (see figures below).

In the above figure, each color represents a different lineage (L1-pink, L2-blue, L3-purple, L4-red, L5-brown, L6-green).